# Cellular geometry and epithelial-mesenchymal plasticity intersect with PIEZO1 in breast cancer cells
Choon Leng So[1,5], Mélanie Robitaille[1], Francisco Sadras[1], Michael H. McCullough[2,6], Michael J. G. Milevskiy[3,4], Geoffrey J. Goodhill [2,7], Sarah J. Roberts-Thomson [1] & Gregory R. Monteith [1] ✉

Differences in shape can be a distinguishing feature between different cell types, but the shape of a cell can also be dynamic. Changes in cell shape are critical when cancer cells escape from the primary tumor and undergo major morphological changes that allow them to squeeze between endothelial cells, enter the vasculature, and metastasize to other areas of the body. A shift from rounded to spindly cellular geometry is a consequence of epithelial-mesenchymal plasticity, which is also associated with changes in gene expression, increased invasiveness, and therapeutic resistance. However, the consequences and functional impacts of cell shape changes and the mechanisms through which they occur are still poorly understood. Here, we demonstrate that altering the morphology of a cell produces a remodeling of calcium influx via the ion channel PIEZO1 and identify PIEZO1 as an inducer of features of epithelial-to-mesenchymal plasticity. Combining automated epifluorescence microscopy and a genetically encoded calcium indicator, we demonstrate that activation of the PIEZO1 force channel with the PIEZO1 agonist, YODA 1, induces features of epithelial-to-mesenchymal plasticity in breast cancer cells. These findings suggest that PIEZO1 is a critical point of convergence between shape-induced changes in cellular signaling and epithelial-mesenchymal plasticity in breast cancer cells.

As a fundamental part of all living things, cell shape is critical to functionality, from the biconcave shape of red blood cells to the long axonal processes of interconnected neurons. However, many cell types also need to remain plastic, able to drastically alter their shape to enable movement through the body and to enact other physiological changes[1–3]. Cells have mechanisms to sense transient alterations in their morphology, such as the mechanosensitive calcium ($Ca^{2+}$) channel PIEZO1. PIEZO1 is critical in various processes, including endothelial cells detecting shear stress and regulation of vascular tone and blood pressure[4,5]. PIEZO1 appears to be important in the cell spreading of HEK293 cells on elongated micropatterned surfaces[6] and the $Ca^{2+}$ increases that occur in CHO-α4WT cells as they are physically confined[7].

When moving from primary tumor sites to metastatic sites, cancer cells must squeeze through blood vessels, and in doing so, they change shape[1,8]. Alterations in cellular geometry can be a response to stimuli, such as inducers of epithelial to mesenchymal plasticity[1,9], but cell geometry can also causally contribute to phenotypic transformations. Confinement of NIH 3T3 fibroblasts to rectangular or circular geometries causes the upregulation of serum response factor and NF-κB target genes, respectively[10]. Similarly, in human mesenchymal stem cells, decreasing micropattern size causes the localization of YAP to change from predominantly nuclear to cytoplasmic[11].

[1]School of Pharmacy, The University of Queensland, Woolloongabba, QLD 4102, Australia. [2]Queensland Brain Institute and School of Mathematics and Physics, The University of Queensland, Brisbane, QLD 4072, Australia. [3]ACRF Cancer Biology and Stem Cells Division, The Walter and Eliza Hall Institute of Medical Research, Melbourne, VIC 3052, Australia. [4]Department of Medical Biology, The University of Melbourne, Parkville, VIC 2010, Australia. . [5]Present address: Department of Biochemistry and Molecular Biology, Johns Hopkins Bloomberg School of Public Health, Johns Hopkins University, Baltimore, MD 21205, USA. [6]Present address: Eccles Institute of Neuroscience, John Curtin School of Medical Research, and School of Computing, ANU College of Engineering and Computer Science, The Australian National University, Canberra ACT 2600, Australia. [7]Present address: Departments of Developmental Biology and Neuroscience, Washington University School of Medicine, St. Louis, MO 63110, USA. ✉e-mail: gregm@uq.edu.au

Although some transcription factors and coactivators, such as YAP/TAZ[11] and MRTF[12], are regulated by cellular geometry, there is a lack of knowledge on how cellular geometry influences more dynamic cellular signaling pathways, such as $Ca^{2+}$ influx. However, studies in cell migration (a process that generally relies on cellular geometric changes)[13–15] implicate a potential intersection between $Ca^{2+}$ signaling and cellular geometry. Here, we investigated how cell shape may alter the "$Ca^{2+}$ signaling toolkit"[16,17] components and the potential intersection with the induction of epithelial to mesenchymal plasticity. We found that changes in cellular architecture resulted in an upregulation of *PIEZO1* and consequently the augmentation of PIEZO1-mediated $Ca^{2+}$ influx. We further identified that PIEZO1

activation induces features of epithelial-to-mesenchymal plasticity in breast cancer cells. Collectively, our results suggest how shape-induced changes in cellular signaling and epithelial-mesenchymal plasticity intersect with the PIEZO1 force channel in breast cancer cells.

## Results

### Changes in gene expression when MCF-7 cells are confined to elongated triangular morphology

To investigate how cell shape may alter the "$Ca^{2+}$ signaling toolkit"[16,17] components and epithelial to mesenchymal plasticity-related genes, we grew MCF-7 cells on unpatterned surfaces. Consistent with the epithelial-

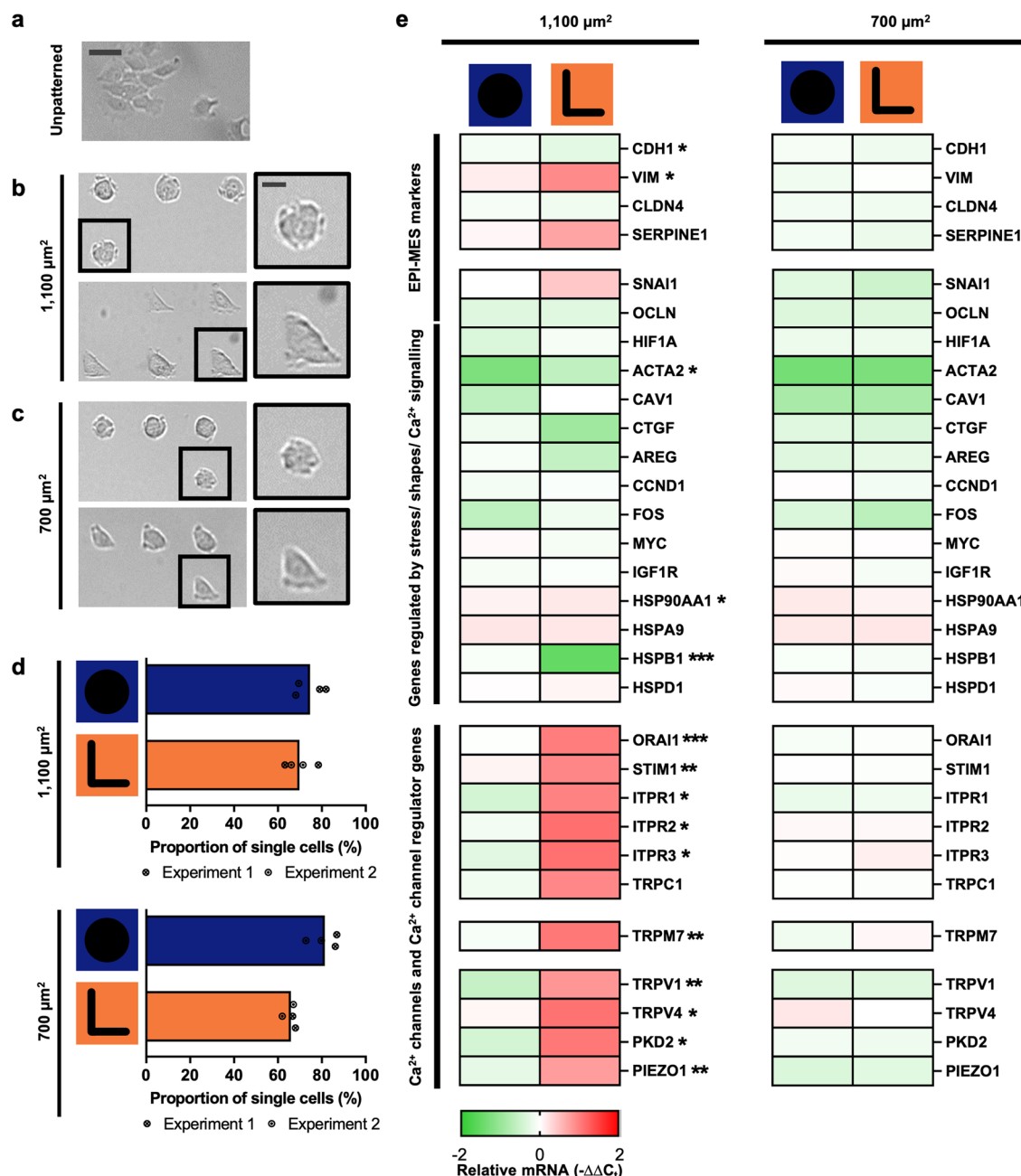

**Fig. 1 | EPI-MES and $Ca^{2+}$ signaling genes are cell geometry dependent in MCF-7 epithelial-like breast cancer cells. a** MCF-7 cells on unpatterned surfaces, **b** 1100 μm² "O" (top) and "L" (bottom) micropatterns, and **c** 700 μm² "O" (top) and "L" (bottom) micropatterns. Scale bar = 50 μm. **d** Proportion of occupied micropatterns with single cells. Data from four individual microplate wells, two independent cell platings: 1100 μm² "O" micropatterns, *n* = 454 micropatterns; "L" micropatterns,

*n* = 390 micropatterns (top); 700 μm² "O" micropatterns, *n* = 508 micropatterns; "L" micropatterns, *n* = 454 micropatterns (bottom). **e** Heatmap of expression changes (-ΔΔC$_t$) including EPI-MES markers, $Ca^{2+}$ channels and $Ca^{2+}$ channel regulators, and other genes. Data represent the average gene expression changes normalized to unpatterned wells (-ΔΔC$_t$) (*n* = 4 biological replicates, except *TRPV1 n* = 3 biological replicates). Unmarked $P \geq 0.05$, *$P < 0.05$, **$P < 0.01$, ***$P < 0.001$ ("O" vs "L").

like features of this breast cancer cell line[18], the cells exhibited an epithelial-like polygonal morphology (Fig. 1a). Using commercially available surface micropatterning (refer to Methods), we then confined MCF-7 cells to various cellular geometry, and we found that the elongated triangular MCF-7 cells ("L" micropattern; 1100 μm²) demonstrated the most pronounced gene expression changes (Supplementary Fig. 1). Figure 1b–d shows the comparison between non-polarized rounded ("O" micropatterned) and elongated triangular ("L" micropatterned) geometries (Fig. 1b–d). Epithelial-mesenchymal (EPI-MES) markers, *VIM* and *CDH1*, differed significantly between rounded MCF-7 cells ("O" micropattern; 1100 μm²) and elongated triangular MCF-7 cells ("L" micropattern; 1100 μm²) (Fig. 1e, left panel). The more elongated triangular MCF-7 morphology was also associated with pronounced upregulation and remodeling of a variety of Ca²⁺ signaling components (Fig. 1e, left panel). Upregulation of gene expression was only observed on larger (1100 μm²), not smaller (700 μm²), micropattern surfaces (Fig. 1e, right panel). The mechanosensitive Ca²⁺ channel, PIEZO1, was one of the Ca²⁺ permeable ion channels significantly upregulated with the elongated triangular geometry (Fig. 1e, left panel).

### Shape-induced alterations in PIEZO1-mediated Ca²⁺ influx

Amongst the Ca²⁺ signaling components that showed pronounced gene upregulation with cellular elongation ("L" micropattern; 1100 μm²), PIEZO1 is a bona fide mechanosensitive Ca²⁺ channel and has roles in the sensing of cellular forces[19,20]. Hence, we next investigated if *PIEZO1* upregulation also corresponded with changes in PIEZO1-mediated Ca²⁺ influx. We compared cytosolic free Ca²⁺ ([Ca²⁺]$_{CYT}$) responses to the pharmacological activator of PIEZO1, YODA 1[21], in rounded and elongated triangular MCF-7 cells ("O" vs "L" micropatterns; 1100 μm², Fig. 2a, b). Assessment of MCF-7 cells expressing the genetically encoded Ca²⁺ sensor, GCaMP6m, via automated epifluorescence microscopy showed that YODA 1 induced rapid, concentration-dependent increases in [Ca²⁺]$_{CYT}$ (Fig. 2c). Differences between the two cellular geometries were most pronounced in the Ca²⁺ influx phase induced by YODA 1, where MCF-7 cells with elongated triangular morphology exhibited a more sustained increase in [Ca²⁺]$_{CYT}$ than rounded MCF-7 cells (Fig. 2b–e). Interestingly, the maximum relative [Ca²⁺]$_{CYT}$ induced by YODA 1 was augmented only at submaximal concentrations in elongated triangular cells (Fig. 2f). This shows that although PIEZO1-mediated Ca²⁺ influx sensitivity is remodeled in elongated MCF-7 breast cancer cells, it is the duration of increases [Ca²⁺]$_{CYT}$ induced by YODA 1 which is the most augmented and can be observed even at maximal concentrations of YODA 1.

We also investigated the underlying low dimensional structure in the time series trajectories by applying principal component analysis (PCA); this further supported augmented PIEZO1-mediated Ca²⁺ influx in the elongated triangular morphologies of MCF-7 cells. More than 95% of the variance in the data can be explained by the top three principal components (PCs, Supplementary Fig. 2a). PC1 and PC2 corresponded to transient responses with relatively slower and faster decay times than the mean of the data, respectively (Fig. 2g). A scatter plot of PC1 and PC2 scores at 3 μM YODA 1 revealed separate clusters for rounded and elongated triangular MCF-7 cells (Fig. 2h). The two clusters were statistically distinct, as revealed by linear discriminant analysis (LDA) (Fig. 2i). On average, the "L" micropattern cluster was characterized by higher values for PC1 and lower values for PC2 relative to the "O" micropattern cluster, indicating slower decay in the elongated triangular cells. This was also evident from the reconstructed trajectories of the cluster means in PCA space (Fig. 2j) and was consistent with enhanced PIEZO1-mediated Ca²⁺ influx in elongated triangular MCF-7 cells.

We further quantified the differences in decay times by fitting Ca²⁺ transients with a nonlinear model (refer to Methods; Fig. 2k, left panel). Figure 2k (middle panel) shows a representative example of a model fit for cells on "O" and "L" micropatterns responding to YODA 1 (3 μM). For data that met the fitting criteria (refer to Methods; Supplementary Fig. 2b), the fitted decay time of the model was significantly longer for cells on "L" micropatterns compared to cells on "O" micropatterns for YODA 1

concentrations at 0.3–3 μM (Fig. 2k, right panel). At these concentrations, the model error was low and not significantly different between "O" and "L" micropatterns (Supplementary Fig. 2c). Collectively, the area under the curve (AUC), PCA, and nonlinear model fitting analyses demonstrated that elongated triangular MCF-7 cells had augmented YODA 1-induced Ca²⁺ influx compared to rounded MCF-7 cells, as well as elevated *PIEZO1* mRNA levels.

### Association between PIEZO1 and EPI-MES markers in breast tumors

Given that selected EPI-MES marker genes were differentially expressed in elongated triangular MCF-7 cells, and that this coincided with a marked increase in *PIEZO1*, we explored these associations in breast tumor samples from the TCGA cohort[22]. Using an extensive list of EPI-MES marker genes[23–25], *PIEZO1* expression clustered with both mesenchymal (*AP1M1*, *SNAI1*) and epithelial (*KRT5*, *KRT7*, *KRT14* and *ST14*) genes (Fig. 3a). Since tumor cells possess a spectrum of EPI-MES phenotypes[26], we developed an EPI-MES signature (Methods; Supplementary Fig. 3a) and stratified all breast tumors from the TCGA breast tumor dataset into quintiles based on their EPI-MES score: high-mesenchymal (H-MES), low-mesenchymal (L-MES), ambiguous (Ambig), low-epithelial (L-EPI) and high-epithelial (H-EPI) (Supplementary Fig. 3b, top panel). When the quintiles were assessed across the PAM50 molecular subtypes, >60% of basal-like breast tumors were assigned H/L-MES, whilst >50% of luminal B tumors were assigned H/L-EPI (Supplementary Fig. 3b, bottom panel). Given the ambiguity of *PIEZO1* expression correlating to both MES and EPI markers, we reassessed correlations within the EPI-MES quintiles (Fig. 3b, Supplementary Fig. 3c). This revealed that both EPI (e.g. *LLGL2*, *CLDN4*, *KRT7*, *ST14*) and MES (e.g. *SNAI1*, *VIM*, *AP1M1*) genes positively correlated with *PIEZO1* expression across all quintiles, whilst markers of luminal breast tumors (*ESR1*, *FOXA1*, *PGR*) negatively correlated in all instances. Of the four marker genes assessed in the morphological analysis, *SERPINE1* displayed an increasing level of correlation as the EPI signature increased in the breast tumors (Fig. 3c). A similar pattern was observed for *ZEB1* (and to a lesser extent *ZEB2*), genes commonly upregulated in mesenchymal cells[27,28]; expression negatively correlated in MES tumors and increased as the EPI signature increased (Fig. 3b). These associations between both EPI and MES genes suggest that *PIEZO1* might be associated with the transition state, rather than definitive epithelial or mesenchymal state.

### PIEZO1 channel activation is associated with EPI-MES plasticity

The link between *PIEZO1*, cellular geometry, and EPI-MES markers could indicate a role for PIEZO1 in EPI-MES plasticity, which is Ca²⁺ influx-dependent in other cell types, including breast cancer cells[29]. Thus we evaluated if augmented PIEZO1-induced Ca²⁺ influx, such as those seen in elongated triangular MCF-7 cells, could induce changes in the expression of the EPI-MES plasticity markers. Our studies were first conducted in MDA-MB-468 breast cancer cells, where the link between EPI-MES plasticity and Ca²⁺ influx has been most directly characterized[29]. YODA 1-mediated Ca²⁺ influx (Supplementary Fig. 4a, b) induced a concentration-dependent decrease in the epithelial marker, E-cadherin, and a concentration-dependent increase in the mesenchymal marker, vimentin, in MDA-MB-468 breast cancer cells (Fig. 4a, b). YODA 1 also affected concentration-dependent changes in the expression of EPI-MES markers, with significant upregulation of *CDH1*, *VIM*, and *CLDN4* in MDA-MB-468. However, the most pronounced increase was seen in the EPI-MES plasticity marker, *SERPINE1* (Fig. 4c), where expression in the TCGA tumor samples additionally correlated with *PIEZO1* in an EPI-MES score-dependent manner (Fig. 3c). *SERPINE1* levels were particularly sensitive to intracellular Ca²⁺ levels and its upregulation by YODA 1 was Ca²⁺ signal-dependent as demonstrated by the effects of intracellular Ca²⁺ chelation with BAPTA-AM (Supplementary Fig. 4c). YODA 1 induced increases in

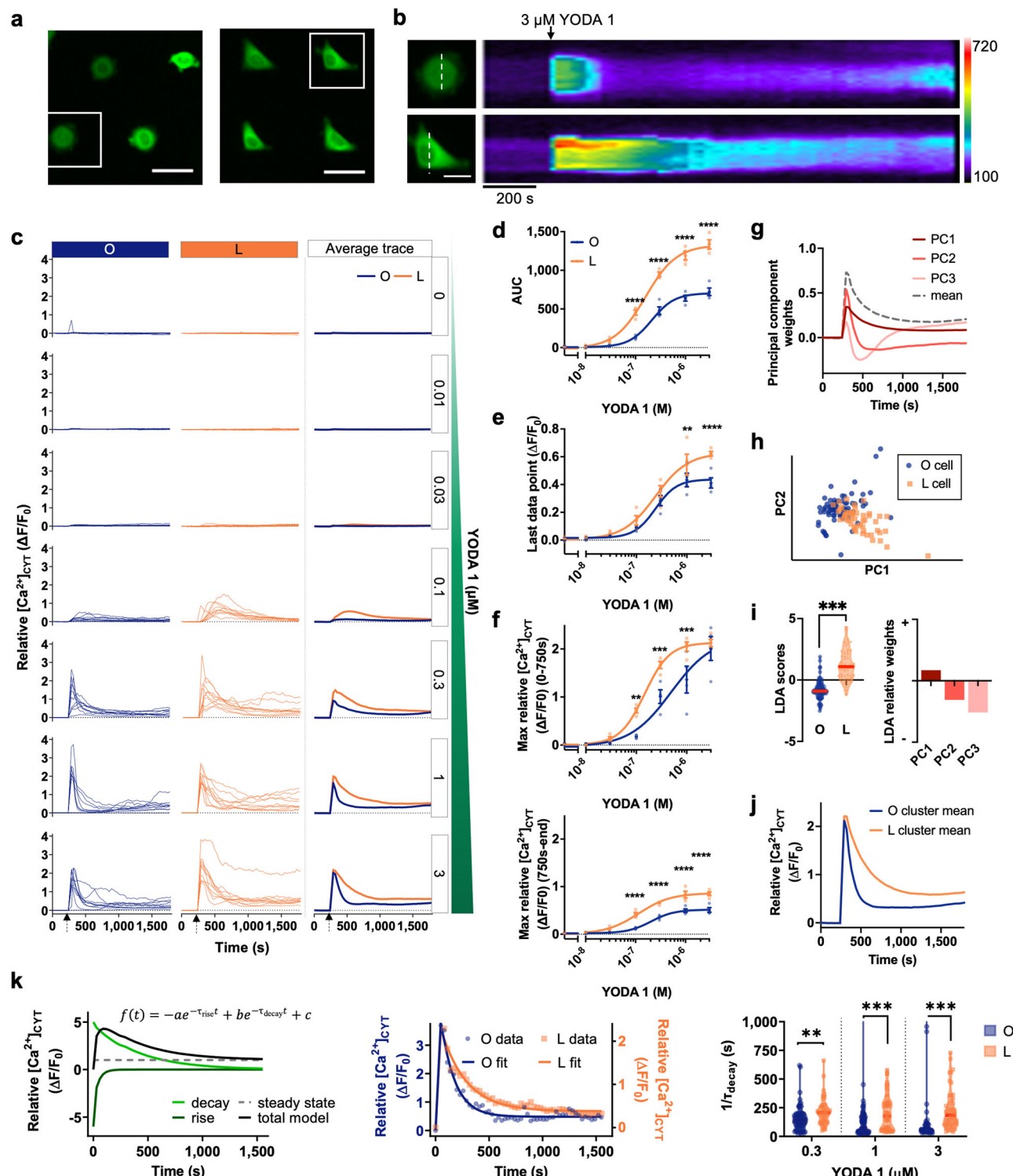

*SERPINE1* were also observed in MCF-7 breast cancer cells (Supplementary Fig. 4d).

To further explore the association between PIEZO1 and EPI-MES markers, we generated a CRISPR/Cas9 *PIEZO1* knockout (KO) in MDA-MB-468 cells with two independent guide RNAs (Fig. 4d, Supplementary Fig. 5a). YODA 1-dependent [Ca$^{2+}$]$_{CYT}$ increases were abolished in the absence of PIEZO1, while ATP and epidermal growth factor (EGF)-induced [Ca$^{2+}$]$_{CYT}$ increases were relatively unaffected (Fig. 4e, f, Supplementary Fig. 6a, b). Basal expression of EPI-MES markers was similar in both scrambled and *PIEZO1*-KO cells (Supplementary Fig. 5b), as was the

induction of EPI-MES markers induced by EGF at the 6 h timepoint (Supplementary Fig. 6c); the latter of which is dependent on Ca$^{2+}$ influx and transient receptor potential-melastatin-like 7 (TRPM7)[29]. However, increases in EPI-MES markers induced by YODA 1, and in particular, *SERPINE1*, were clearly attenuated in the *PIEZO1*-KO cells and in cells with transient *PIEZO1* silencing using siRNA (Fig. 4g, Supplementary Fig. 7). YODA 1-induced changes in vimentin and E-cadherin protein were moderately reversed in the *PIEZO1*-KO cells, suggesting some YODA 1 off-target effects and highlighting the importance of our silencing studies (Supplementary Fig. 5c, d). Collectively, these studies link PIEZO1

**Fig. 2 | Ca²⁺ influx induced by the PIEZO1 activator, YODA 1, in MCF-7 cells is dependent on cellular geometry. a** Representative fluorescence images of GCaMP6m-MCF-7 cells on "O" (left) and "L" (right) micropatterns. Scale bar = 50 μm. **b** Line scan (dotted line, scale bar = 20 μm) of YODA 1 (3 μM) induced changes in relative $[Ca^{2+}]_{CYT}$ (GCaMP6m fluorescence intensity in arbitrary units shown in pseudocolour; scale bar = 200 s) over time in cells on "O" (top) and "L" micro-patterns (bottom). **c** $[Ca^{2+}]_{CYT}$ changes in GCaMP6m-MCF-7 cells on "O" and "L" micropatterns with increasing YODA 1 concentrations. $[Ca^{2+}]_{CYT}$ changes in ten randomly selected representative cells on "O" (left) and "L" (middle) micropatterns and averages of all cells (right). **d** $[Ca^{2+}]_{CYT}$ area under the curve (AUC), **e** sustained $[Ca^{2+}]_{CYT}$, last data point (ΔF/F₀) and **f** maximum relative $[Ca^{2+}]_{CYT}$ change (ΔF/F₀) for cells on "O" and "L" micropatterns induced by increasing concentrations of YODA 1 (0–3 μM). From 0–750 s (top panel) and 750 s –end (bottom panel). Data shown represent the mean ± S.E.M. **g** Top components from PCA captured

differences between slow and fast decay. **h** Scatter plots of PC1 and PC2 scores of single cells activated by 3 μM YODA 1. **i** Means of "O" and "L" micropatterned cell clusters were significantly different at 3 μM YODA 1 (left). Linear discriminant analysis (LDA) shows that cluster means were best separated along a direction corresponding to the difference between fast and slow decay (right; i.e., positive LDA weight for PC1 and negative weight for PC2). **j** Reconstructed Ca²⁺ transients for cluster means show that "L" micropatterned cells decayed more slowly than "O" micropatterned cells at 3 μM YODA 1. **k** Illustration of a nonlinear model for Ca²⁺ transients incorporating an exponential rise, a slower exponential decay, and a steady-state constant (left). Representative model fits (median model error) for "O" and "L" micropatterned cells, respectively, at 3 μM YODA 1 (middle). Fitted decay time ($1/\tau_{decay}$) was significantly longer for "L"-shaped cells at 0.3, 1, and 3 μM YODA 1 (right; Y-axis is truncated to $1/\tau_{decay}$ = 1000 s) (n = 4 biological replicates). Unmarked $P \geq 0.05$, **$P < 0.01$, ***$P < 0.001$, ****$P < 0.0001$.

activation to the process of EPI-MES plasticity via modulation of selected EPI-MES markers.

## PIEZO1 activation induces the localization-reset of YAP1 and upregulates SERPINE1 via YAP1

PIEZO1 modulates the localization and activity of YAP1, a mechanosensitive transcriptional coactivator[30]. YODA 1 activation in MDA-MB-468 breast cancer cells induced a YAP1 localization reset phenomenon. This was characterized by initial depletion and subsequent accumulation of nuclear YAP1 (Fig. 5a, b), which is essential for YAP1 transcriptional activity in MCF10A breast cells[31]. Here, we also observed a decrease in nuclear YAP1 upon YODA1 treatment at 15 min, and a subsequent "reset" at 1 h time point (Fig. 5a, b). YODA 1-induced mRNA upregulation of the YAP1 target gene, CTGF, was sensitive to YAP1 silencing, as was SERPINE1 (Fig. 5c, Supplementary Fig. 8a, b), which is also a downstream YAP1 gene[32,33]. Expression of SERPINE1 and CTGF was additionally sensitive to silencing of TEAD4, a well-documented transcription factor that complexes with YAP1[34] (Fig. 5c). While the YAP1 paralog, TAZ (encoded by WWTR1), regulated the expression of CTGF, the expression of SERPINE1 was not TAZ-dependent in MDA-MB-468 cells (Fig. 5c). Silencing components of the mechanosensitive serum response factor (SRF) transcription pathway (MKL1, MKL2, and SRF) attenuated YODA 1-induced upregulation of CTGF, as expected[35]; however, these components were also not involved in YODA 1-induced upregulation of SERPINE1 (Supplementary Fig. 8c–e).

## Discussion

Using surface micropatterning to induce a change in cellular geometry, we showed alterations in gene transcription including genes that are key markers of EPI-MES and those involved in Ca²⁺ signaling. We identified that PIEZO1, a force-activated Ca²⁺ channel, is upregulated with an elongated triangular cellular morphology and this is associated with an augmentation of intracellular Ca²⁺ increases induced by the PIEZO1 activator, YODA 1. The most pronounced gene expression changes were observed in the 1100 μm² elongated triangular but not at 700 μm², when compared to other shapes. The maximum distance between two points was 25% greater for 1100 μm² surfaces than for 700 μm² micropattern surfaces: 37.4 μm compared to 29.9 μm ("O" micropattern) and 66.3 μm compared to 52.9 μm ("L" micropattern). Given that MCF-7 cells had a surface area of ~1017 μm² (from n = 448 segmented MCF-7 cells on unpatterned surface), the 1100 μm² provides the most suitable surface area for assessment. The elongated triangular cellular geometry creates a long single free-edge organization not seen in other micropatterns used in this study. These features could influence the expression and activity of the bona fide mechanosensitive Ca²⁺ channel, PIEZO1[19,20].

The augmentation of intracellular Ca²⁺ increases induced by the PIEZO1 activator, YODA 1, in 1100 μm² "L" cells could be explained by the upregulation of PIEZO1 in these cells. Other potential mechanisms could include the sensitivity, kinetics, or spatial distribution of PIEZO1, or changes in other associated Ca²⁺ signaling pathways. Indeed,

Yang et al. found that the membrane curvature can regulate the spatial distribution of Piezo1 on the plasma membrane[36]. As previous studies have found that micropatterning can create localized tension at specific subcellular regions[37], future studies could investigate PIEZO1 protein expression and the spatial distribution of PIEZO1 in cells adopting different cellular geometries and their correlation with local tensions at subcellular regions. Future work could also evaluate the relative contributions of changes in PIEZO1 levels and/or channel kinetics to the shape-dependent increases in calcium influx induced by YODA 1 that have been observed in our studies.

Various forms of mechanical stimuli can remodel Ca²⁺ signaling. PIEZO1 is associated with cellular organization and upregulation of PIEZO1 occurs in disorganized malignant T4-2 breast cells compared with non-malignant S1 cells that grow in organized acini[38]. Mechanical stress (in the form of cardiac hypertrophy) similarly results in the induction of PIEZO1 expression and augments electrically evoked Ca²⁺ transients in mouse cardiomyocytes[39], and membrane curvature regulates the plasma membrane distribution of PIEZO1[36]. Prior to the identification of PIEZO1, Lee et al. observed that fish epithelial keratocytes with innate fibroblastic shape demonstrate more frequent Ca²⁺ transients than fan-shaped cells[40]. Our results show that in addition to being a force sensor[19,20] and regulator of cellular morphology in CHO-α4WT cells[7] and muscle stem cells[41], PIEZO1 can be remodeled as a consequence of changes in cell shape, further exemplifying the interdependence and dynamic reciprocity of mechanical signaling.

We and others have shown that Ca²⁺ signaling influences EPI-MES plasticity in breast cancer cells[29,42]. We now show that the activation of PIEZO1 caused pronounced induction of the EPI-MES marker SERPINE1 in breast cancer cells. This result along with a previous study in MC3T3-E1 osteoblasts[43], suggests that SERPINE1 gene expression is very sensitive to PIEZO1 activation. SERPINE1 (which encodes for PAI1) is a prognostic marker of poor clinical outcomes in breast cancer patients[44,45], has pro-tumorigenic roles[46], and is upregulated when triple-negative breast cancer cells transition from epithelial to a mesenchymal state[47]. We also showed that SERPINE1 induction by PIEZO1 activation was dependent on the transcriptional coactivator YAP1, which is a well-characterized influencer of cellular proliferation and apoptosis[48].

One feature of EPI-MES plasticity is the loss or decrease of junctional protein E-cadherin[1]. In MDA-MB-468 breast cancer cells, YODA 1 activation resulted in an increase in vimentin and a decrease in E-cadherin. The association of PIEZO1 and junctional proteins is also seen through the interaction of Piezo1 with PECAM1[49], and Piezo1-induced remodeling of junctional proteins in endothelial cells and lymphatic endothelial cells[50,51]. Future studies could further evaluate the intersection of PIEZO1 and junctional proteins in the context of cancer progression pathways.

Ca²⁺ signaling plays a critical role in normal cellular processes[16,17,52] and in many disease processes[4,5,53,54]. Cancer cells change morphology while metastasizing to sites distant from the primary tumor and Ca²⁺ signaling is critical in processes important in cancer progression, such as proliferation and cell migration[54]. The results presented here

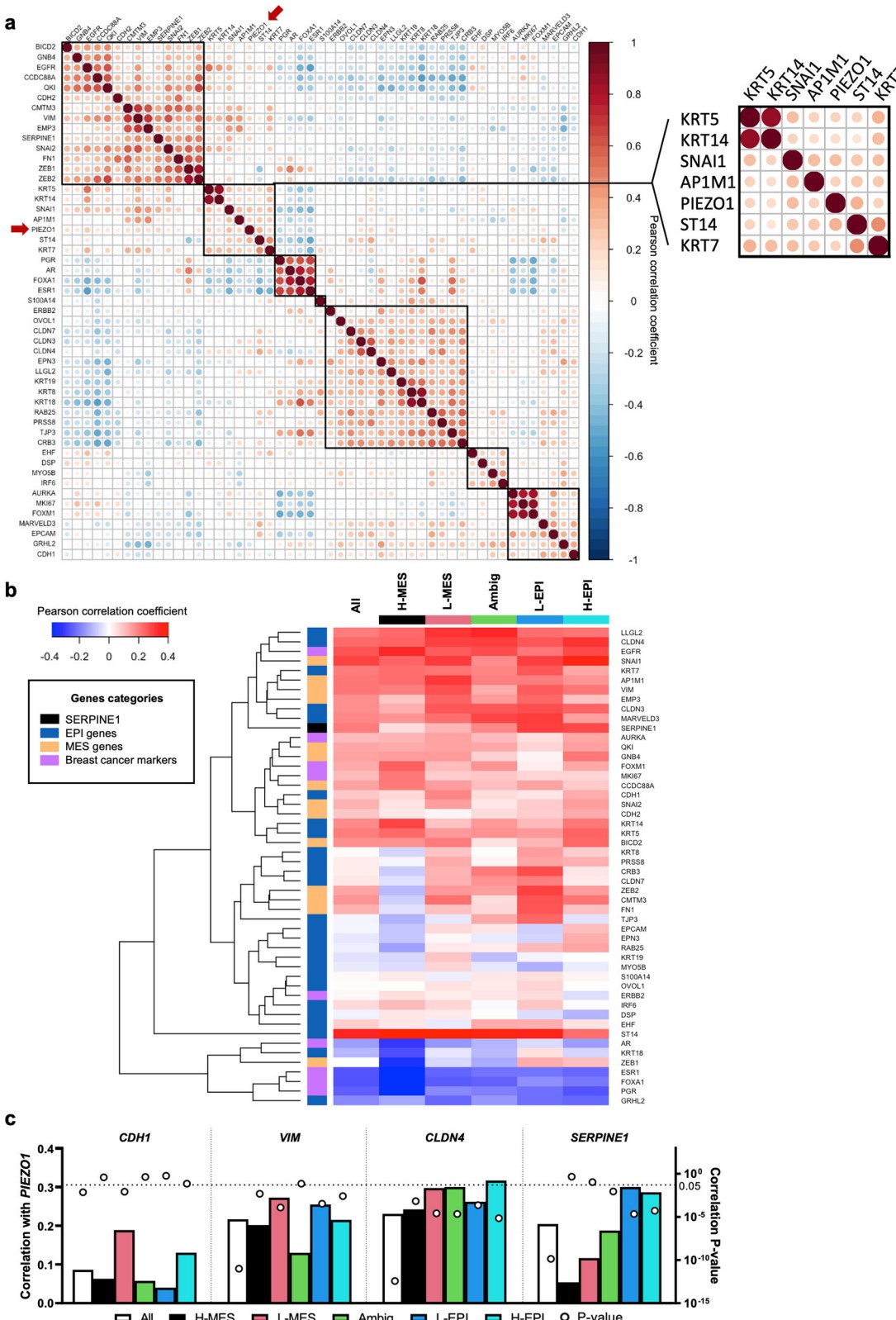

**Fig. 3 | *PIEZO1* expression correlates with specific EPI-MES markers in breast tumours. a** Correlation matrix for mRNA expression of *PIEZO1* and the EPI, MES, and breast cancer marker genes in the TCGA breast tumour dataset. The inset shows genes grouped with *PIEZO1* by hierarchical clustering. **b** Correlation of *PIEZO1* expression with EPI, MES, and breast cancer marker genes in each EPI-MES quintile: high-mesenchymal (H-MES), low-mesenchymal (L-MES), ambiguous (Ambig), low-epithelial (L-EPI) and high-epithelial (H-EPI). **c** Correlation of *PIEZO1* with *CDH1, VIM, CLDN4* and *SERPINE1* in each EPI-MES quintile. The left Y-axis is the Pearson correlation coefficient of each gene with *PIEZO1*. The right Y-axis is the correlation P-value. The horizontal dotted line represents *P* = 0.05.

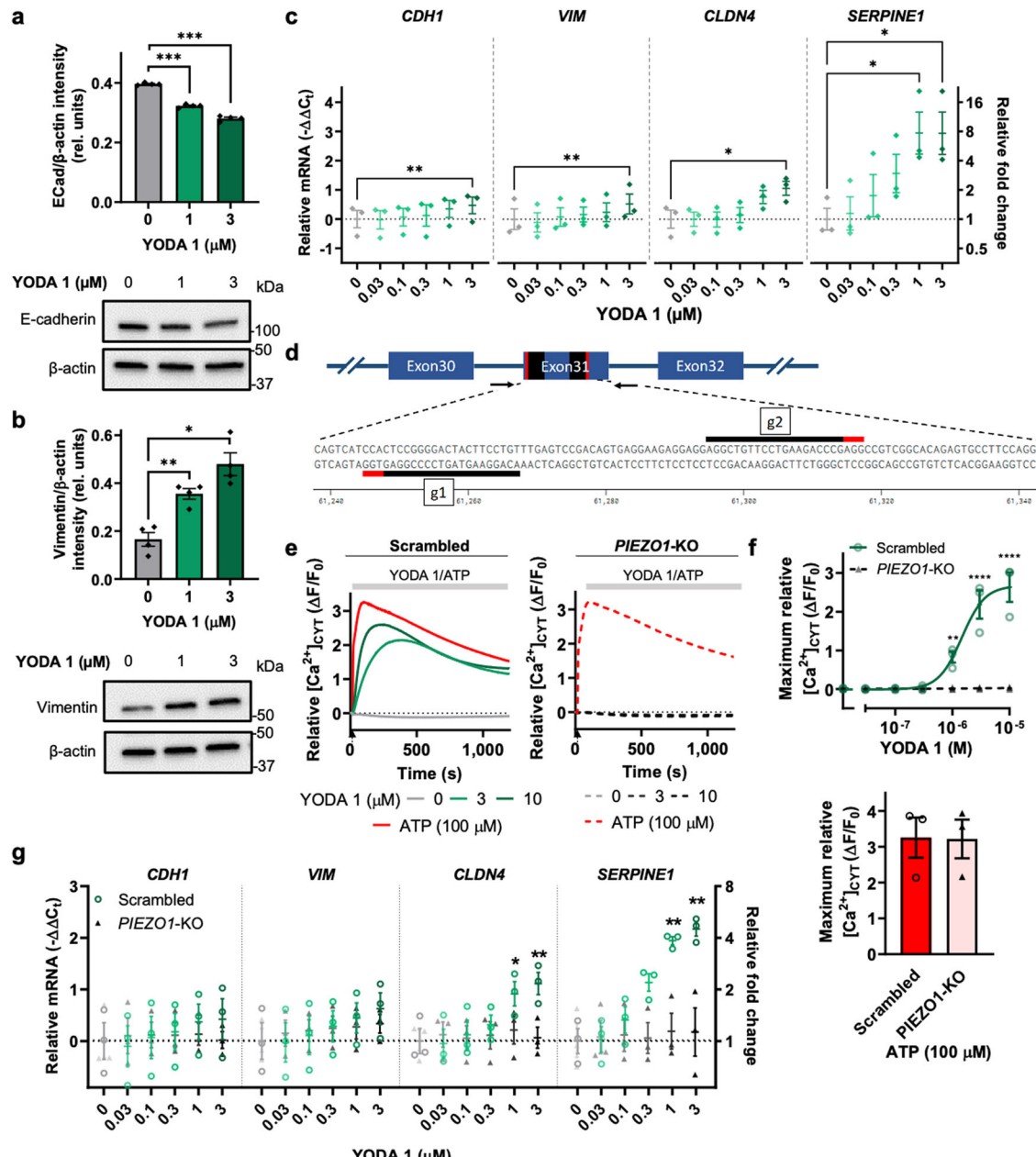

**Fig. 4 | The PIEZO1 activator, YODA 1, induces changes in the expression of specific EPI-MES markers in breast cancer cells. a** Densitometry and representative immunoblot showing downregulation of the epithelial marker E-cadherin in MDA-MB-468 by YODA 1 (72 h). **b** Densitometry and representative immunoblot showing YODA 1 induction of the mesenchymal marker, vimentin, in MDA-MB-468 cells (24 h). β-actin was used as the loading control. The data shown represent the mean ± S.E.M. ($n$ = 4 biological replicates). **c** *CDH1, VIM, CLDN4* and *SERPINE1* mRNA levels in MDA-MB-468 cells stimulated with YODA 1 (3 h). The data shown represent the mean ± S.E.M. ($n$ = 3 biological replicates). **d** Schematic representation of the strategy for *PIEZO1*-KO, gRNA sequences (g1, g2) in black, PAM sites in red. Arrows represent the positions of the designed primers for genomic PCR. **e** Knockdown of *PIEZO1* eliminates YODA 1 but not ATP-induced increases in $[Ca^{2+}]_{CYT}$ in MDA-MB-468 breast cancer cells; mean relative $[Ca^{2+}]_{CYT}$ ($\Delta F/F_0$) from duplicate wells of three independent experiments. **f** Maximum relative $[Ca^{2+}]_{CYT}$ ($\Delta F/F_0$) induced by YODA 1 (0–10 µM) (top) and 100 µM ATP (bottom). **g** *CDH1, VIM, CLDN4* and *SERPINE1* mRNA levels in MDA-MB-468-*PIEZO1*-KO cells and MDA-MB-468-scrambled cells stimulated with YODA 1 (3 h). Where appropriate, the data shown represent the mean ± S.E.M. ($n$ = 3 biological replicates). Unmarked $P \geq 0.05$, *$P < 0.05$, **$P < 0.01$, ***$P < 0.001$, ****$P < 0.0001$.

demonstrate a clear intersection between cellular geometry, $Ca^{2+}$ signaling, the PIEZO1 force channel, and EPI-MES plasticity in breast cancer cells. Our identification that some $Ca^{2+}$ signaling pathways become more predominant with an elongated triangular morphological cellular state, will pave the way for the improved targeting of specific cellular processes important in different diseases. The high-throughput and automated approaches used to define cell morphology-dependent alterations in $Ca^{2+}$ signaling represent a powerful new approach to identifying agents and targets that could disrupt the interplay between cellular geometry, cell signaling pathways, and cellular plasticity.

## Methods
### Cell lines and cell culture
MCF-7 and HEK293T cell lines were sourced from ATCC. The MDA-MB-468 cell line was from The Brisbane Breast Bank, UQCCR, Australia. All cell lines were cultured in DMEM (Sigma-Aldrich) supplemented with 10% v/v

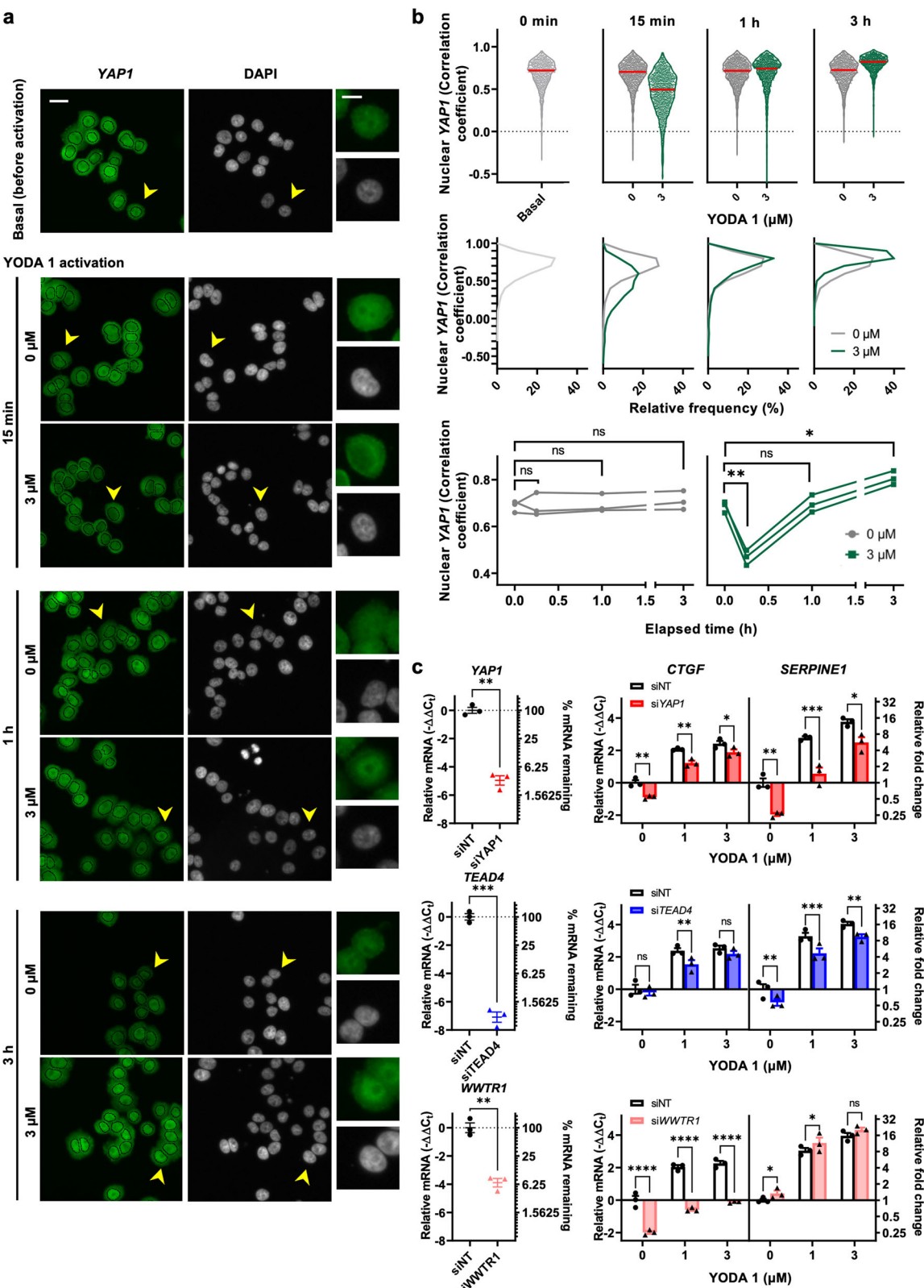

**Fig. 5 | YODA 1 induces localization-reset of YAP1 and upregulates YAP1 target gene SERPINE1 in breast cancer cells. a** Representative images showing immunofluorescence of YAP1 with dotted lines indicating the nuclei (left) and nuclei (DAPI, middle) of MDA-MB-468 cells at 0 min (basal), 15 min, 1 h and 3 h following 0 μM and 3 μM YODA 1 stimulation. Scale bar = 20 μm. Zoomed-in images showing YAP1 localization and the nucleus of a representative cell (right, scale bar = 10 μm). **b** YODA 1 induced an initial depletion (15 min) and a subsequent accumulation of nuclear YAP1. Nuclear YAP1 changes quantitated as correlation coefficient, represented as violin plots (top), histograms (middle), and means (bottom). **c** mRNA expression of *CTGF* and *SERPINE1* in MDA-MB-468 cells after ON-TARGETplus SMARTpool siRNA-mediated silencing of *YAP1*, *TEAD4* and *WWTR1* (encodes for *TAZ*). The data shown represent the mean ± S.E.M. (*n* = 3 biological replicates). ns *P* ≥ 0.05, *\**P* < 0.05, *\*\**P* < 0.01, *\*\*\**P* < 0.001, *\*\*\*\**P* < 0.0001.

FBS (Life Technologies) and 4 mM L-glutamine (Thermo Fisher Scientific) for less than 10 passages post-thaw in a humidified 37 °C incubator with 5% $CO_2$. Unless otherwise indicated, experiments conducted in serum-reduced conditions were in FluoroBrite DMEM (Thermo Fisher Scientific) supplemented with 0.5% v/v FBS, 4 mM L-glutamine, and 25 mM HEPES buffer (Thermo Fisher Scientific). All cell lines were routinely tested for mycoplasma contamination. MCF-7 and MDA-MB-468 cell lines were routinely authenticated by short tandem repeat (STR) profiling. To assess YODA 1-induced mRNA expression changes, cells were serum-reduced for 24 h, detached using Accumax (Invitrogen), and seeded into 96-well plates (3000 cells per well for MCF-7 and 6000 cells per well for MDA-MB-468) in serum-reduced media for 48 h before being treated with YODA 1 (Tocris) for the duration stated in the figure legends. To assess YODA 1-induced protein expression changes, YODA 1-induced YAP1 translocation and hEGF (Sigma-Aldrich)-induced mRNA expression changes, MDA-MB-468 cells were seeded into 6-well plates (250,000 cells/well) or 96-well plates (6000 cells/well) for 24 h before being serum-reduced for 24 h. Cells were then treated with YODA 1 or hEGF for the duration stated in the figure legends. To assess the effect of BAPTA-AM on YODA 1-induced mRNA expression changes, MDA-MB-468 cells were seeded into 96-well plates (6000 cells/well) for 24 h before being serum-reduced for 48 h, Cells were then treated with 100 µM BAPTA-AM (B6769, Invitrogen) or DMSO for 1 h, before YODA 1 treatment.

### Cell culture on micropatterned plates
CYTOOplates 96 Starter-A (CYTOO) were used for live-cell imaging and qRT-PCR studies. MCF-7 cells were serum-reduced for 48 h, detached using 0.2% w/v EDTA in pH 7.4 PBS/EDTA, resuspended and pelleted twice with media, and syringed slowly through a 23 G needle four times prior to plating at a density of 3,000 cells per well using serum-reduced media. Wells were washed twice 1 h after plating to remove unattached cells. Cells were allowed to attach for 24 h before performing experiments. For the $[Ca^{2+}]_{CYT}$ studies, GCaMP6m-MCF-7 cells were detached using Accumax and plated on 1100 µm² "O" micropatterned (CYTOOplate 96 RW DC-M-A, CYTOO) and 1100 µm² "L" micropatterned (CYTOOplate 96 RW L-M-A, CYTOO) plates using a similar protocol. Unless otherwise indicated, cells were allowed to attach for 16–24 h before performing experiments.

### Brightfield imaging
Brightfield imaging was performed using the brightfield mode of a JuLi Stage microscope (NanoEntek) using a 4× objective kept in a humidified 37 °C incubator with 5% $CO_2$. To determine the proportion of single-cells, images at the 24-h timepoint were exported into ImageJ, and cells were manually counted using the Cell Counter plugin[55].

### RNA preparation and qRT-PCR
RNA was isolated using an RNeasy Plus Micro Kit (Qiagen) or RNeasy Plus Mini Kit (Qiagen) according to the manufacturer's protocol. Multiple wells (four to six) were pooled together to provide sufficient mRNA yield for qRT-PCR of multiple targets. cDNA was synthesized using the Omniscript Reverse Transcription Kit (Qiagen) according to the manufacturer's protocol. RNase inhibitor (Promega) and random primers (Promega) were purchased separately.

qRT-PCR was performed using TaqMan Gene expression assays (Applied Biosystems) with TaqMan® Fast Universal PCR Master Mix (2×), no AmpErase® UNG (Life Technologies) in a StepOne Plus Real-Time PCR System (Applied Biosystems). Gene expression changes induced by micropatterning were normalized to the endogenous control 18S rRNA. Unless otherwise indicated, other target gene expression were normalized to PUM1. Quantitation was performed using the comparative $C_t$ (threshold cycle) method[56]. Unless specified, statistical analyses were performed on log2 transformed relative fold change (relative mRNA, $-\Delta\Delta C_t$). TaqMan gene expression assays used were 18S 4319413E, ACTA2 Hs00426835_g1, AREG Hs00950669_m1, CAV1 Hs00971716_m1, CCND1 Hs00765553_m1, CLDN4 Hs00976831_s1, CDH1 Hs00170423_m1, CTGF

Hs00170014_m1, FOS Hs00170630_m1, HIF1A Hs00153153_m1, HSP90AA1 Hs00743767_sH, HSPB1 Hs03044127_g1, HSPA9 Hs00269818_m1, HSPD1 Hs01036753_g1, IGF1R Hs00609566_m1, ITPR1 Hs00181881_m1, ITPR2 Hs00181916_m1, ITPR3 Hs01573555_m1, MKL1 Hs01090249_g1, MKL2 Hs00401867_m1, MYC Hs00153408_m1, OCLN Hs00170162_m1, ORAI1 Hs03046013_m1, PIEZO1 Hs00207230_m1, PKD2 Hs00960946_m1, PUM1 Hs00472881_m1, SERPINE1 Hs01126606_m1, SNAI1 Hs00195591_m1, SRF Hs00182371_m1, STIM1 Hs00162394_m1, TRPC1 Hs00608195_m1, TRPM7 Hs00292383_m1, TRPV1 Hs00218912_m1, TRPV4 Hs01099348_m1, VIM Hs00185584_m1, WWTR1 Hs00210007_m1, and YAP1 Hs00902712_g1.

### Lentiviral vectors, lentivirus production and infection
GCaMP6m was amplified from pGP-CMV-GCaMP6m (a gift from Dr. Douglas Kim, Addgene plasmid #40754) and cloned into pCDH-EF1-FHC lentiviral vector (a gift from Dr. Richard Wood, Addgene plasmid #64874) as previously described[57]. CRISPR/Cas9 plasmids were constructed from lentiCRISPRv2 hygro (a gift from Dr. Brett Stringer, Addgene plasmid #98291). Short guide sequences (sgPIEZO1#1: ACAGGAAG-TAGTCCCCGGAG, sgPIEZO1#2: AGGCTGTTCCTGAAGACCCG, scrambled#1: GCACTACCAGAGCTAACTCA and scrambled#2: CCTAAGGTTAAGTCGCCCTCG) were cloned between the BsmBI restriction sites using a protocol deposited by the Zhang Lab on Addgene[58]. All CRISPR/Cas9 plasmids were sequenced with Sanger sequencing using forward primer for human U6 promoter.

Lentiviral particles were produced by transfection using Lipofectamine 2000 (Invitrogen) on HEK293T cells, as previously described[59]. To generate GCaMP6m-MCF-7 cells, MCF-7 cells were infected for 24 h in the presence of 8 µg/mL polybrene (Merck). Selection with 2 µg/mL puromycin (Sigma) was applied 48 h after infection. Mixed populations of GCaMP6m-MCF-7 cells were fluorescence-activated cell sorted (FACS) into single cells using Beckman Coulter MoFlo Astrios EQ (Translational Research Institute, Australia), and viable clones were then expanded and the expression of hormone receptors at mRNA level, expression of epithelial markers at protein level and STR profiling were used to validate the selected clone.

To generate MDA-MB-468-PIEZO1-KO and MDA-MB-468-scrambled cells, MDA-MB-468 cells were transduced twice with viral media (equal volumes of sgPIEZO1#1 and sgPIEZO1#2 or equal volumes of scrambled#1 and scrambled#2, respectively) in the presence of 8 µg/mL polybrene (48 h per infection). Selection with 400 µg/mL hygromycin B (Gibco) was applied 48 h after the second infection. Selected cells were used immediately as a mixed population for less than 10 passages and were not cryopreserved. Functional knockdown of PIEZO1 was assessed with each experiment/passage using YODA 1-induced $[Ca^{2+}]_{CYT}$ increases in MDA-MB-468-PIEZO1-KO cells and MDA-MB-468-scrambled cells (Fig. 4e, f) and genotyping (Methods and Supplementary Table 1).

### Transient gene silencing
MDA-MB-468 cells were seeded into 96-well plates (6000 cells per well) for 24 h before siRNA transfection. Cells were then incubated in 8% v/v FBS transfection media containing DharmaFECT4 (0.1 µL per well, Dharmacon) and 100 nM of siRNA for 24 h. The following SMARTpool ON-TARGETplus siRNAs (Dharmacon) were used: Non-Targeting Pool (D-001810-10), MKL1 (57591) siRNA (L-015434-00), MKL2 (57496) siRNA (L-019279-00), PIEZO1 (9780) siRNA (L-020870-03), SRF (6722) siRNA (L-009800-00), TEAD4 (7004) siRNA (L-019570-00), WWTR1 (25937) siRNA (L-016083-00), and YAP1 (10413) siRNA (L-012200-00). The following siGENOME SMARTpool siRNAs (Dharmacon) were used: Non-Targeting Pool #2 (D-001206-14-05) and YAP1 (10413) siRNA (M-012200-00). The transfection media was removed and changed to serum-reduced media for 48 h before experiments. Gene silencing was confirmed using qRT-PCR. Statistical analysis was performed on relative mRNA ($-\Delta\Delta C_t$). The % mRNA remaining was calculated as relative fold change ($2^{-\Delta\Delta Ct}$) × 100%.

## High-content imaging of intracellular Ca²⁺ fluorescence using automated epifluorescence microscopy

An ImageXpress Micro automated epifluorescence microscope was used to assess $[Ca^{2+}]_{CYT}$ in single GCaMP6m-MCF-7 cells on micropatterned plates (100 μL FluoroBrite DMEM, supplemented with 0.5% v/v FBS, 4 mM L-glutamine and 25 mM HEPES buffer). Images were acquired with a 10× objective an excitation filter of 472/30 nm and an emission filter of 520/35 nm with an exposure time of 40 ms for GCaMP6m every 30 s for 30 min (referred to as calcium images). YODA 1 solutions were prepared in FluoroBrite DMEM, supplemented with 0.5% v/v FBS, 4 mM L-glutamine, and 25 mM HEPES buffer to obtain the final concentrations as stated. At the tenth time point, 20 μL of the prepared YODA 1 was added to the micropatterned plates. Images were continually acquired before, during, and after YODA 1 addition.

At the end of the experiment, nuclei were counterstained with Hoechst 33342 dye, and images were acquired at two wavelengths: (1) 40 ms exposure with a 10× objective and an excitation filter of 472/30 nm and an emission filter of 520/35 nm for GCaMP6m, and (2) 5 ms exposure with a 10× objective and an excitation filter of 377/50 and emission filter of 447/60 nm for Hoechst 33342 dye (referred to as post-Hoechst images). Unless otherwise specified, all images were acquired with binning at $3 \times 3$ to improve the signal-to-noise ratio of each image. Experiments were performed across four independent cell passages, in duplicate wells per treatment across two independent ImageXpress time-lapse experiments (two independent cell passages each time-lapse experiment).

Image analysis was performed using MetaXpress High Content Image Acquisition and Analysis software using a custom-written analysis journal. Data from each part of the journal analysis (described below) were exported in CSV format as individual data files. All data files were subsequently combined and analyzed using R (refer to Supplementary Fig. 9 for a general overview).

The first image from the raw time-lapse calcium images was removed and not further analyzed. The remaining 60 images were auto-aligned using the first plane as the reference plane. From the aligned time-lapse calcium images, an image stack of the first image was segmented to identify regions containing GCaMP6m-MCF-7 cells. Image segmentation was performed using the auto-find object tracking algorithm with an adaptive threshold. The settings for minimum width, maximum width, and intensity above the local background were 10 pixels, 60 pixels, and 5 intensity values, respectively. Segmented objects were eroded by 3 pixels to account for any slight image drift between timepoints. This created a mask that could identify a single cell as a region and allowed for tracking of $[Ca^{2+}]_{CYT}$ changes (GCaMP6m emission intensity) for each single cell over time (referred to as calcium GCaMP6m object, Data file #1).

Post-Hoechst nuclei images were used to obtain single-cell information. The post-Hoechst nuclei image was duplicated to create an image stack, which was segmented to identify regions containing nuclei. Image segmentation was performed using the auto-find object tracking algorithm with an adaptive threshold. The settings for minimum width, maximum width, and intensity above the local background were 10 pixels, 20 pixels, and 800 intensity values, respectively. Segmented nuclei were binarized and dilated by 3 pixels. Objects touching the border were removed. The remaining objects were then eroded back by 3 pixels to create a new Hoechst mask.

Each Hoechst-tracked object on the new Hoechst mask was processed in two ways: a) associate each nucleus (from the Hoechst mask) to individual cells (GCaMP6m object), and b) identify regions that contained only a single cell. First, to associate each Hoechst object to each GCaMP6m object, the Hoechst mask was overlaid on the post-Hoechst GCaMP6m image to match each Hoechst-tracked object with post-Hoechst GCaMP6m object (Data file #2). Second, to identify regions that contained only single cells, the minimum distance of each Hoechst-tracked object to every other Hoechst-tracked object was obtained; a close distance would indicate more than one cell (nuclei) per region. To do that, an Euclidean distance image for each Hoechst-tracked object was generated. Using the Euclidean distance image, the minimum distance of each Hoechst-tracked object to every other Hoechst-tracked object was identified and logged (Data file #3). Finally, for each cell, their post-Hoechst GCaMP6m object was associated with the corresponding calcium GCaMP6m object. This was achieved by aligning and matching the post-Hoechst GCaMP6m image and the first image from the time-lapse calcium images as part of the analysis journal (Data file #4).

Files were processed using R (version 3.5.1)[60]. Briefly, the data files (#1 – 4) from each well were combined and merged. Raw data were filtered so that only data from single cells that adopted the intended shapes were assessed. For the "O" shape, GCaMP6m objects with shape factors of ≥0.9 were included (a shape factor of 1 is a perfect circle). For the "L" shape, a GCaMP6m object with a shape factor of 0.5–0.85 (inclusive) was included. As each micropattern was sometimes occupied by more than one cell, only micropatterned regions that contained a single nucleus (reflective of one cell attached to a region) were included in the analysis. Data from regions with only one cell were identified, by identifying nuclei (Hoechst-tracked object) that were at least 20 pixels away from other nuclei. Large nuclei could also indicate more than one nucleus per region. Therefore, only nuclei that were less than 300 μm² (the size of the largest single nuclei based on manual sampling) were included in the data analysis. To ensure that the GCaMP6m objects (calcium GCaMP6m object and post-Hoechst GCaMP6m object) and the Hoechst-tracked object were matched correctly, the centroid locations from all files were compared. Data were excluded if the differences of any two sets of centroids were > 30 μm—the approximate width of one cell. Data that fulfilled the above criteria were analyzed. Unless otherwise stated, for pooled quantitative data, data points are the mean of duplicate wells across four independent cell passages assessed as two time-lapse ImageXpress experiments (two independent cell passages each time-lapse experiment).

## $[Ca^{2+}]_{CYT}$ imaging using fluorometric imaging plate reader (FLIPR)

$[Ca^{2+}]_{CYT}$ changes in MDA-MB-468 cells in response to YODA 1, ATP (Sigma-Aldrich) and hEGF were assessed using a fluorometric imaging plate reader FLIPR$^{TETRA}$ (Molecular Devices) using the PBX no-wash Ca²⁺ Assay Kit (BD Biosciences), as previously described[29] in serum-reduced FluoroBrite DMEM or physiological salt solution (PSS; 10 mM HEPES, 5.9 mM KCl, 1.4 mM MgCl₂, 1.2 mM NaH₂PO₄, 5 mM NaHCO₃, 140 mM NaCl, 11.5 mM glucose) with 1.8 mM CaCl₂ (Sigma-Aldrich) or 100 μM BAPTA tetrasodium salt, cell impermeant (Invitrogen). Unless otherwise stated, YODA 1, ATP, and hEGF were prepared on a compound plate. FLIPR$^{TETRA}$ was programmed to automatically add 50 μL to the cell-containing sample plates to obtain the final concentrations as stated in the figures and Ca²⁺ signaling changes were continuously assessed before, during, and after reagent addition.

## Protein isolation and immunoblot

Cells were lysed using protein lysis buffer containing PhosSTOP (Sigma Aldrich) and cOmplete™ Mini Protease Inhibitor Cocktail (Sigma Aldrich). Gel electrophoresis was performed using 4-15% Mini-PROTEAN TGX Stain-Free Protein Gel (Bio-Rad) and transferred to polyvinylidene difluoride membranes (Trans-Blot Turbo Mini 0.2 μm PVDF Transfer Packs, Bio-Rad). The membrane was blocked for 1 h in 5% w/v skim milk powder in PBS containing 0.1% v/v Tween-20 (Aldrich Chemical Company) before treating with primary antibodies overnight at 4 °C. Membrane was then treated with horseradish peroxidase-conjugated secondary antibodies for 1 h at room temperature before being imaged using SuperSignal West Dura Extended Duration Substrate (Thermo Scientific) and Chemi-Doc Imaging System (Bio-Rad). Each band was quantified using Image Lab Software (version 5.2.1, Bio-Rad) or ImageJ[55]. Densitometry was performed relative to the β-actin loading control. The antibodies used were anti-PIEZO1 antibody (1:1000 in 5% w/v milk PBST, NBP275617, Novus Biologicals), anti-vimentin antibody (1:1,000 in 5% w/v milk PBST, V6389, Sigma), anti-E-cadherin antibody (1:1,000 in 5% w/v milk PBST, 14472, Cell

Signaling Technology), anti-β-actin antibodies (1:10,000 in 5% w/v milk PBST, A5441, Sigma-Aldrich) and horseradish peroxidase-conjugated secondary goat-anti-mouse antibodies (1:10,000 in 5% w/v milk PBST, 170-6516, Bio-Rad).

## Immunofluorescence

Cells were fixed with formaldehyde solution, methanol free (4% w/v, Thermo Scientific) for 30 min, permeabilized and blocked using PBS containing 10% v/v goat serum (Sigma Aldrich), 0.3 M glycine (Sigma Aldrich), 1% w/v bovine serum albumin (Sigma Aldrich) and 0.1% v/v Triton X-100 (Sigma Aldrich) for 1 h before treating with anti-YAP1 antibody (63.7) (1:200 in blocking buffer, SC-101199, Santa Cruz) overnight at 4 °C. Cells were then treated with goat-anti-mouse Alexa 488 antibody (1:200 in blocking buffer, #4408 S, Cell Signaling) for 1 h at room temperature. Nuclear staining was performed using DAPI (1 μg/mL, Invitrogen) incubated at room temperature for 10 min before imaging using ImageXpress Micro at: (1) 100 ms exposure with an excitation filter of 472/30 nm and an emission filter of 520/35 nm for GFP, and (2) 2 ms exposure with an excitation filter of 377/50 and an emission filter of 447/60 nm for DAPI, with a 20× objective and binning at 2 × 2. Analysis of nuclear YAP1 was performed using the Standard Translocation application in MetaXpress High Content Image Acquisition and Analysis software. The settings for minimum width and intensity above the local background were 10 pixels and 50 intensity values, respectively.

## Breast tumor correlation analysis

Genes used to explore the epithelial or mesenchymal state of breast tumors were sourced from ref [23–25]. Epithelial genes were: *CDH1, CLDN3, CLDN4, CLDN7, CRB3, DSP, EHF, EpCAM, EPN3, GRHL2, IRF6, KRT5, KRT5, KRT8, KRT14, KRT18, KRT19, LLGL2, MARVELD3, MYO5B, OVOL1, PRSS8, RAB25, S100A14, ST14* and *TJP3*. Mesenchymal genes were: *AP1M1, BICD2, CCDC88A, CDH2, CMTM3, EMP3, FN1, GNB4, QKI, SERPINE1, SNAI1, SNAI2, VIM, ZEB1* and *ZEB2*. Additionally, markers of breast cancer molecular subtypes were also included: *AR, AURKA, EGFR, ERBB2, ESR1, FOXA1, FOXM1, MKI67*, and *PGR*. Expression values (RSEM) were obtained from the TCGA[22] online data portal, cBioPortal[61,62], and log2 transformed for analysis. Pearson correlation analysis was done using the rcorr function in the corrplot package[63] of R v4.1.2. Hierarchical clustering was achieved via average-linkage of the Manhattan distance; rectangles were assigned based on the results of the hierarchical clustering, seven for all tumors in Fig. 3a and six for each of the MES/EPI groups in Supplementary Fig. 3c. To establish refined signatures for MES and EPI genes, the Z-scores of all genes from both lists were correlated together across all breast tumors, those genes displaying a Pearson correlation coefficient > 0.5 to the average Z-score of all MES or EPI genes were used to make the final signatures: MES (*CCDC88A, CMTM3, FN1, QKI, SNAI2, VIM, ZEB1* and *ZEB2*) and EPI (*CLDN4, CRB3, KRT8, KRT18, PRSS8* and *RAB25*). Finally, the MES Z-score of each tumor was subtracted from the EPI Z-score to then give a relative EPI-MES score, where negative values indicate a tumor with a higher level of MES gene expression and positive scores a higher EPI (Supplementary Fig. 3a).

## Principal component analysis of transient calcium response to YODA 1 agonist

The fluorescence data for each cell $i$ are scalar time series $\boldsymbol{y}_i = \{y_n\}_{n=0}^N$, where $y_n$ is the detrended and normalized fluorescence value at the $n^{th}$ time point for a total of $N$ observations. Data were recorded with a consistent sampling rate and were time-aligned with respect to the addition of the YODA 1 agonist. Therefore, to investigate patterns in the shape of the transient calcium responses, we treated each $\boldsymbol{y}_i$ as a single vectorial observation with $N$ dimensions and applied principal component analysis (PCA). Components were computed by using Scikit-learn[64], with a full singular value decomposition solver for the complete data pooled over "O" and "L" cell shapes, and for all tested concentrations of the agonist. We reduced the dimensionality of the data by retaining the three components that explained

the most variance. We refer to the projection onto these three components as PC space (Fig. 2g and h, and Supplementary Fig. 2a).

We applied a MANOVA to observations in PC space using Statsmodels[65] to determine if the means of the clusters corresponding to each cell shape were different (Fig. 2i). The test was applied to the data for each agonist concentration level separately and *p*-values were adjusted for multiple comparisons by Bonferroni correction. To understand the nature of the identified differences between the calcium responses for different cell shapes, we applied linear discriminant analysis (LDA) to the same observations in PC space using Scikit-learn[64]. The weights of the linear projection learned by LDA were used to interpret shape-related differences. We visualized the estimated average calcium responses corresponding to the means of the respective clusters for each cell shape in PC space by projecting the cluster centroids back into the full $N$-dimensional space of $\boldsymbol{y}_i$ (Fig. 2j).

## Nonlinear model fitting

Data were fitted using nonlinear least squares optimization with a Trust Region Reflective algorithm[66], as implemented in the SciPy function *optimize.curve_fit*[65].

We modelled transient calcium responses as an exponential rise and decay process given by:

$$f(t) = -ae^{-\tau_{\text{rise}}t} + be^{-\tau_{\text{decay}}t} + c, \tag{1}$$

where $f(t)$ is the detrended and normalized fluorescence time series, $t$ is time, $\tau_{\text{rise}}$ and $a$ are constants controlling the rate and magnitude of exponential rise, respectively, $\tau_{\text{decay}}$ and $b$ are constants controlling the rate and magnitude of exponential decay, respectively, and $c$ is the constant corresponding to the steady state of the system (Fig. 2k, left panel). To ensure that the model conformed to a rise followed by a decay (not a decay followed by a rise), we constrained $\tau_{\text{decay}} < \tau_{\text{rise}}$. To achieve this we defined:

$$\tau_{\text{rise}} \triangleq \tau_{\text{decay}} + \tau_\Delta, \tag{2}$$

where $\tau_\Delta \geq 0$ and replaces $\tau_{\text{rise}}$ as a model parameter, and rewrote the model as:

$$f(t) = -ae^{-\left(\tau_{\text{decay}} + \tau_\Delta\right)t} + be^{-\tau_{\text{decay}}t} + c. \tag{3}$$

To force the model to start with zero fluorescence at the time that the transient began (i.e., $f(0) = 0$) we further constrained the model by:

$$a = b + c, \tag{4}$$

and rewrote the final model with four parameters as:

$$f(t) = -(b + c)e^{-\left(\tau_{\text{decay}} + \tau_\Delta\right)t} + be^{-\tau_{\text{decay}}t} + c. \tag{5}$$

We used nonlinear least squares optimization with a Trust Region Reflective algorithm[66] as implemented in the SciPy function *curve_fit*[67] to fit the model in Eq. 5. Data were fitted to each time series $\boldsymbol{y}_i' = \{y_n\}_{n=8}^N - \bar{y}_{\text{baseline}}$, where $\bar{y}_{\text{baseline}}$ is the mean of $\{y_n\}_{n=0}^7$ and $n = 8$ is the time point corresponding to the start of the calcium transient. We specified additional bounds on the parameters during fitting. Firstly, to ensure that the fit conformed to the desired shape of a rise followed by a decay, we constrained $\tau_{\text{decay}} \geq 0$, $\tau_\Delta \geq 0$, and $b \geq 0$. Secondly, we assumed that the steady state must be greater than or equal to zero and less than the maximum fluorescence value observed (after the baseline adjustment in $\boldsymbol{y}_i'$). Hence, $0 \leq c \leq \max \boldsymbol{y}_i'$. It follows that $a \geq b$, which is also necessary to maintain the desired model behavior.

Model parameters must be initialized when fitting using the Trust Region Reflective algorithm. We initialized the steady-state constant $c$ with

the median of $\mathbf{y}_i'$, which was typically a good approximation of this parameter for our data. Using the median was more effective than the mean because the data were positively skewed due to the asymmetrical nature of the calcium transient. For the other parameters, $\tau_{\text{decay}}$, $\tau_\Delta$ and $b$, we ran the model fit many times for a range of initializations and selected the fit with the lowest $r^2$ error (sum of squared residuals). This was to avoid local minima in the solution space. Specifically, we used a 3-dimensional grid search for the best nonlinear fit over ten logarithmically spaced values of $\tau_{\text{decay}}$, such that $\frac{1}{\tau_{\text{decay}}}$ ranged from 15 to 1500 seconds, six logarithmically spaced values of $\tau_\Delta$ from $10^{-6}$ to $10^{-1}$, and five logarithmically spaced values of $b$ from 0.1 to 1000.

We only attempted to fit the model to the data for the $i$th cell if $\mathbf{y}_i'$ met three specified criteria. Firstly, the median of $\mathbf{y}_i'$ had to be greater than or equal to zero, or else $c$ would be initialized outside of the parameter bounds. Secondly, the distribution of the observations $\{y_n\}_{n=8}^N$ (i.e., the transient response) had to have higher fluorescence values on average than those for $\{y_n\}_{n=0}^7$ (i.e., prior to the transient response), as determined by a one-sided Wilcoxon rank-sum test with significance level $\alpha = 0.005$, otherwise we deemed the cell to be non-responsive. Thirdly, the maximum of $\mathbf{y}_i'$ had to occur in the first half of the time series. If this condition was not met, and given that the model specifies that transients must decay more slowly than they rise, then the fit of $\tau_{\text{decay}}$ might not be reliable because the modelled transient would not be able to converge to the steady state value over the observed period (Supplementary Fig. 2b).

### Genotyping MDA-MB-468-PIEZO1 KO and MDA-MB-468-scrambled cells

Knockdown of *PIEZO1* was confirmed by genotyping (Supplementary Table 1). The genomic DNA of two passages of MDA-MB-468-*PIEZO1* KO and MDA-MB-468-scrambled cells (passage 6 and 9) were extracted using a GeneJET Genomic DNA Purification Kit (Thermo Scientific) and amplified using a Phusion High-Fidelity PCR Kit (Thermo Scientific) with forward primer 5'-CCACCTCCCAGGTTCAAAGG-3' and reverse primer 5'-CAGATCCAGATCCCTGCTCTG-3'. Amplicons were purified using agarose gel electrophoresis and the GeneJET Gel Extraction Kit (Thermo Scientific). Purified amplicons were sequenced with Sanger sequencing (Australian Genome Research Facility, Australia). Deconvolution analysis was performed using ICE Synthego software.

### Statistics and reproducibility

PCA and nonlinear model fitting analyses[68], were performed with Python in Jupyter Lab using NumPy[69], SciPy[67], Scikit-learn[64], and Statsmodels[65]. Unless otherwise indicated, data were visualized and analyzed using R[60] or GraphPad Prism (version 7.01 or 9.0 for Windows). Unless otherwise indicated, statistical analysis was performed using a two-tailed paired t-test or two-way ANOVA with Šidák's multiple comparisons test with the matched pairing of each cell passage/experiment. For Fig. 2i (left panel), statistical analysis was performed using MANOVA (Wilk's lambda (F statistic) with 3 degrees of freedom; Bonferroni corrected for multiple comparisons with respect to different YODA 1 concentrations). For Fig. 2k (right panel) and Supplementary Fig. 2c, statistical analysis was performed using two-sided Wilcoxon rank-sum tests (Fig. 2k, right panel was Bonferroni corrected for multiple comparisons with respect to different YODA 1 concentrations). For Figs. 4a–c, 5b (bottom panel), and Supplementary Fig. 5c, statistical analysis was performed using two-way ANOVA with Dunnett's multiple comparisons test with the matched pairing of each cell passage/experiment. Experiments on micropatterned plates were from four independent cell passages on two independent cell platings for qRT-PCR or two independent time-lapse ImageXpress experiments. Other data were from at least three independent experiments. Unless otherwise indicated, where appropriate, n indicates the number of independent experiments.

### Reporting summary

Further information on research design is available in the Nature Portfolio Reporting Summary linked to this article.

## Data availability

Uncropped and unedited blot images are in Supplementary Fig. 10. All data supporting the findings of this study are available within the paper as figures and source data are available within Supplementary Data 1–5. All other data are available from the corresponding author upon reasonable request.

## Code availability

The custom code used for PCA and nonlinear model fitting analyses is available through GitHub (https://github.com/mhmcc/calcium-PIEZO1) and deposited at https://doi.org/10.5281/zenodo.10828300.

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

## Acknowledgements
This research was partly supported by the National Health and Medical Research Council of Australia (1181922 and 1196855). We thank Mr. Stéphane Guillou for assistance with the R codes used for data analysis. We thank Dr. Blake Chapman for the manuscript writing feedback.

## Author contributions
C.L.S., S.J.R-T, and G.R.M. conceived the study and designed the research. C.L.S. and M.R. performed the experiments. C.L.S., F.S., and M.H.M. analyzed the $Ca^{2+}$ signaling data for micropatterned cells. M.J.G.M. performed bioinformatics analyses. C.L.S. performed all other data analyses. G.J.G., S.J.R-T. and G.R.M. provided supervision and assisted with data interpretation. C.L.S. and G.R.M. wrote the manuscript with input from all authors.

## Competing interests
The authors declare no competing interests.
