## [Peer Review File · Communications Biology]

Reviewers' comments:

Reviewer #1 (Remarks to the Author):

Summary:

Here the authors have demonstrated that changes in geometric pattern (shape) from circle to a triangle can trigger upregulation of mesenchymal genes due to increased calcium signaling in a Piezo-1 dependent manner. Using genetic and drug perturbations, the authors have further showed that Piezo-1 mediated calcium signaling can alter Yap localization and upregulation of EPI-MES markers.

Comments:

I think this is an interesting observation and I want to commend the authors for conducting a detailed investigation into role of calcium signaling in epi-mes transition. I enjoyed reading the manuscript and learned a few things along the way. I think the authors have done robust statistical analysis of calcium signaling and correlated it with gene expression. Especially, I find the correlation between shape and epi-mes transition fascinating and intriguing.

However, I have certain misgivings regarding the experimental methods and interpretation of the results. I think the authors need to address this to improve the quality of the manuscript and its conclusions.

1. It is unclear to me why the authors arbitrarily decide on a circular and L shaped pattern. Maybe I missed something here, but it would be good to give an explanation to the readers about the rationale of the geometric patterns. For instance, if the objective was to compare elongated vs rounded shaped cells, the authors could have designed a rectangular pattern.
2. Changes in geometric pattern from O to L is not equivalent to membrane stretching as the surface area is still the same. Also why did the authors chose only two areas- 700 and 1100 μm^2 ? Nothing interesting happens in both O and L-shaped patterns with area of 700 μm^2 , the authors should do similar experiment a pattern larger than 1100 μm^2 . This will indicate if there is anything special about 1100 μm^2 or the geometry plays a significant role only in stretched out cells. It is interesting why in smaller area there is no difference in gene expression between O and L shaped patterns.
3. To claim that indeed the increased calcium spikes are due to stretching and not geometric patterns, the authors should perform a cell stretching experiment. This will help decouple cellular stretch from geometric configuration, which might be a perturbation on its own. To do this experiment, I suggest the authors can use a custom built or off-the-shelf cell stretcher.
4. The experiments with YODA1 are interesting. However, it is not clear what the reaction kinetics of yoda1 in these cells is. It's unclear in these cells, when was YODA1 added and how much time does it take for it to change cellular behavior. Cells can have calcium spikes even when media is changed. Here, the authors need to do a control with changing media without yoda1 as a control. Any mechanical perturbation including changing media will cause a calcium spike.
5. The authors need to present a longitudinal plot indicating the short-term and longer- term calcium transients before and after YODA1 administration.
6. It will be valuable to plot the frequency of calcium transients with and without yoda 1, rather than the amplitude. The steady state calcium transients are missing in these plots, which is a common phenotype

in Gcamp6 based calcium imaging.

7. Coming back to the reaction kinetics of yoda1, it will be valuable to image for piezo1 channel using immunofluorescence. I am not convinced that administration of Yoda1 will instantaneously increase the localization and activity of piezo1 channel. Also, doing so may help demonstrate if shape has a role in the localization of piezo1 on the plasma membrane.

In conclusion, I think the authors need to revise the manuscript to address the mentioned concerns. This I believe will help strengthen the conclusions drawn by the authors.

Reviewer #2 (Remarks to the Author):

The manuscript by So et al identifies that changing cell morphology can modify Piezo1 expression and calcium signal that induces features of EPI-MES plasticity. The manuscript is interesting and suitable to the scope of Communications Biology. My main concerns are on the first part of the manuscript where the authors establish the correlation between cell geometry and Piezo1 expression. I also have some concern on the interpretation of the Ca signal and the Yap1 signal.

1. The main claims are based on two types of cell shapes (O and L) on 1100um² substrates. It's hard to generalize and claim that cell shape regulates Piezo1 expression, especially that measurements on 700 um² do not seem to follow the same trend as on 1100um² substrates. More substrate sizes and shapes are needed to support the authors' claim.
2. does cell morphology alter local tensions in subcellular regions?
3. what are the size of cells when they are not constrained to the patterned substrate? Are the cells stretched or compressed in the authors experiments?
4. In figure 2, is the difference in Ca influx explained by different Piezo1 expression? If not, what are the molecular reasons that caused the different Ca dynamics? There is also an odd long-term increase of Ca signal on O cells.
5. Is the Piezo1-induced changes in marker expression solely a Ca effect? Can these changes be induced by increasing intracellular Ca²⁺ level while not activating Piezo1?
6. In figure 5a, the changes in Yap1 level is very unclear.

Reviewer #3 (Remarks to the Author):

In this manuscript, the authors explore the potential role of the mechanosensitive ion channel PIEZO1 in cell shape and plasticity. In addition to investigating the link between Ca²⁺ signalling, cell shape and epithelial to mesenchymal plasticity. Of particular interest is the link between PIEZO1 and SERPINE1 expression and other genes involved in epithelial to mesenchymal plasticity. The role of PIEZO1 in cell morphology is an emerging area and the work presented is novel and of interest to the field. The manuscript overall is very well written and presented.

Major comments

1. Are protein levels of Piezo1 changed in o versus L shaped cells? please provide western blots of protein expression levels for MCF 7 cells cultured in O or L conformations, if technically feasible.
2. Are the channel kinetics affected by changes in cell shape or is the increased calcium influx due to increased protein expression? Please provide evidence using patch clamp.
3. Does PIEZO1 knockdown abrogate changes in cell morphology in MCF7 cell cultured in O and L morphologies? The group have used the necessary CRISPR/Cas9 PIEZO1 knockout technology or transient knockout capabilities in this manuscript.
4. Is PIEZO1 sensitivity to mechanical stimulation (as opposed to pharmacological stimulation) affected during changes in cell morphology in the O and L morphologies?
5. Please confirm changes in PIEZO1 protein expression in MDA-MB-468 cells as well as in knockdowns.

Minor Comments

6. The work proposes that the increase in PIEZO1 activity is due to increased gene expression over the 16-24 hour culture period. How does this culture time scale compare to physiological relevant changes in cell shape? Please provide more description of the physiological relevance of the O/L pattern culture system.
7. Nonomura et al, Friedrich et al and Chuntharpursat-Bon et al, have previously reported changes in VE cadherin/B-actin upon YODA1 treatment in endothelial cells. The significance e-cadherin should be discussed in more detail and how it relates to physiological epithelial remodelling.
8. The interpretation of the results should be in a separate discussion section and should go into more detail for each conclusion.

Reviewers' comments:

Reviewer #1 (Remarks to the Author):

Summary:

Here the authors have demonstrated that changes in geometric pattern (shape) from circle to a triangle can trigger upregulation of mesenchymal genes due to increased calcium signaling in a Piezo-1 dependent manner. Using genetic and drug perturbations, the authors have further showed that Piezo-1 mediated calcium signaling can alter Yap localization and upregulation of EPI-MES markers.

Comments:

I think this is an interesting observation and I want to commend the authors for conducting a detailed investigation into role of calcium signaling in epi-mes transition. I enjoyed reading the manuscript and learned a few things along the way. I think the authors have done robust statistical analysis of calcium signaling and correlated it with gene expression. Especially, I find the correlation between shape and epi-mes transition fascinating and intriguing.

However, I have certain misgivings regarding the experimental methods and interpretation of the results. I think the authors need to address this to improve the quality of the manuscript and its conclusions.

We thank the reviewer for their positive and encouraging comments and are glad they enjoyed reading our manuscript. We are excited to further improve the paper based on this reviewer's suggestions.

1. It is unclear to me why the authors arbitrarily decide on a circular and L shaped pattern. Maybe I missed something here, but it would be good to give an explanation to the readers about the rationale of the geometric patterns. For instance, if the objective was to compare elongated vs rounded shaped cells, the authors could have designed a rectangular pattern.

We thank the reviewer for raising this point. We conducted initial qRT-PCR studies in many shapes (CYTOOplates 96 Starter-A, commercially available), and we have now included this early data in the manuscript to provide more context (Supplementary Fig. 1), which showed that gene expression changes were more pronounced in 1100 μm^2 L, which was the reason why we focused on this shape for further study. L is known to create a single free-edge organization (see figure below from the CYTOO website <https://cytoo.com/micropattern-products/plates/cytooplates%E2%84%A2-96-rw-starter>), showing L represents a single free edge organization). This detail has also now been added to the manuscript (page 2 lines 78 – 82 and Supplementary Fig. 1).

A whole range of micropatterns for diverse applications

	Disc	Crossbow	H	Y	L
Micropatterns					Cells					Description	No polarization	Strong polarization	Symmetric organization	Triaxial symmetry organization	Single free edge organization
Noteworthy Applications	Cell arraying Ciliogenesis	Cell polarity Organelle positioning Receptor internalization	Cell division Cell-cell junction	Multipolar division	Cytoskeleton rearrangement & Spindle orientation

Supplementary Fig. 1

919
920
921
922
923
924
925
926

Supplementary Fig. 1: 1,100 μm^2 "L" micropatterns changes the expression of EPI-MES and Ca^{2+} signalling genes in MCF-7 epithelial like breast cancer cells. Heatmap of expression changes ($-\Delta\Delta\text{C}_t$) including EPI-MES markers, Ca^{2+} channels and Ca^{2+} channel regulators and other genes on 1,100 μm^2 (left) or 700 μm^2 (right) micropatterned plates. Data represent the average gene expression changes normalized to unpatterned wells ($-\Delta\Delta\text{C}_t$) (n = 4 biological replicates, except *TRPV1* n = 3 biological replicates). Unmarked $P \geq 0.05$, * $P < 0.05$, ** $P < 0.01$, *** $P < 0.001$ ("O" vs "L").

"Using commercially available surface micropatterning (refer to Materials and Methods), we then confined MCF-7 cells to various cellular geometry, and we found that the elongated triangular MCF-7 cells ("L" micropattern; 1,100 μm^2) demonstrated the most pronounced gene expression changes (Supplementary Fig. 1). Fig. 1b-d shows the comparison between non-polarized rounded ("O" micropatterned) and elongated triangular ("L" micropatterned) geometries (Fig. 1b-d)." (page 2 lines 78 – 82)

2. Changes in geometric pattern from O to L is not equivalent to membrane stretching as the surface area is still the same. Also why did the authors chose only two areas- 700 and 1100 μm^2 ? Nothing interesting happens in both O and L-shaped patterns with area of 700 μm^2 , the authors should do similar experiment a pattern larger than 1100 μm^2 . This will indicate if there is anything special about 1100 μm^2 or the geometry plays a significant role only in stretched out cells. It is interesting why in smaller area there is no difference in gene expression between O and L shaped patterns.

In terms of 700 and 1100 μm^2 , we now note in the manuscript that the 1100 μm^2 is closer to the surface area of MCF-7 cells where the shape is not confined see page 5 lines 215 – 217 (~1017 μm^2 , from 448

segmented MCF-7 cells). The larger surface area of micropatterns (1,600 μm^2) would not be biologically relevant for MCF-7 cells. We chose 700 and 1100 μm^2 due to our use of high-content imaging equipment; we were limited to the use of commercially available cell surface pattern microplates (CYTOOplates 96 Starter-A) – the larger surface area micropatterns from the company are only available as one full 96-well plate for one micropattern (1600 μm^2) – this posed a limitation for our initial aim of screening for “interesting” shapes.

We agree with the reviewer that the term “stretching” for L vs. O at 1100 μm^2 lacks sufficient clarity and could cause confusion, so we no longer use this term. We agree with the reviewer (as reflected in our original title), that it is shape / cellular geometry that is the feature which drives our reported changes. In this context, we have added as a supplementary figure data from other shapes showing that the gene expression changes are unique to “L”. This is now discussed in detail in lines 212 – 215 “The most pronounced gene expression changes were observed in the 1,100 μm^2 elongated triangular but not at 700 μm^2 , when compared to other shapes. The maximum distance between two points was 25% greater for 1,100 μm^2 surfaces than for 700 μm^2 micropattern surfaces: 37.4 μm compared to 29.9 μm (“O” micropattern) and 66.3 μm compared to 52.9 μm (“L” micropattern).” and shown in Supplementary Fig. 1. The L micropattern is known to create a single free-edge organization; hence we have changed from using the term “stretched” to elongated triangular geometry or single free-edge organization.

3. To claim that indeed the increased calcium spikes are due to stretching and not geometric patterns, the authors should perform a cell stretching experiment. This will help decouple cellular stretch from geometric configuration, which might be a perturbation on its own. To do this experiment, I suggest the authors can use a custom built or off-the-shelf cell stretcher.

As discussed above we have now removed the term stretching from the paper. In a high throughput calcium imaging instrument that was required to fully define the calcium signaling changes observed in our work, it would not be possible to use a custom-built or off-the-shelf cell stretcher to dynamically alter cell geometry. Such an advancement would be a transformative breakthrough in the field and is beyond the scope of this paper.

4. The experiments with YODA1 are interesting. However, it is not clear what the reaction kinetics of yoda1 in these cells is. It's unclear in these cells, when was YODA1 added and how much time does it take for it to change cellular behavior. Cells can have calcium spikes even when media is changed. Here, the authors need to do a control with changing media without yoda1 as a control. Any mechanical perturbation including changing media will cause a calcium spike.

We thank the reviewer for their comments and we are happy to clarify and add further details regarding our Ca^{2+} measurement protocols. In all the calcium experiments (Figs. 2, 4, and Supplementary Figs. 2, 4, 6), YODA1 was added using a custom-created protocol on our automated high-content microscope (IXM) or automatically by our high-throughput system (FLIPR). In brief, cells were plated in 96-well sample plates (100 μL media), and DMSO control or YODA1 (six times or three times the final desired concentrations for IXM and FLIPR, respectively) was diluted using media in separate 96-well compound plates. The drugs were aspirated from the compound plates (20 μL and 50 μL for IXM and FLIPR, respectively), added to the sample plate, and the calcium signaling changes were continuously recorded. Therefore, the addition of DMSO (control) also represented the control for any mechanical perturbation as a consequence of liquid addition.

We have now added arrows (below the x-axis) on all our figures to show where control (media addition) and/or YODA or ATP were added, and we have changed the text on pages 8 and 9 (Materials and Methods) to make this clearer.

“YODA 1 solutions were prepared in FluoroBrite DMEM, supplemented with 0.5% v/v FBS, 4 mM L-glutamine, and 25 mM HEPES buffer to obtain the final concentrations as stated. At the tenth time point, 20 μL of the prepared YODA 1 was added to the micropatterned plates. Images were continually acquired before, during and after YODA 1 addition.” (page 8 lines 386 – 390)

“Unless otherwise stated, YODA 1, ATP and hEGF were prepared on a compound plate. FLIPR^{TETRA} was

programmed to automatically add 50 μL to the cell-containing sample plates to obtain the final concentrations as stated in figures and Ca^{2+} signaling changes were continuously assessed before, during and after reagent addition.” (page 9 lines 462 – 465)

Fig. 2

848

849

850

851

852

853

854

855

856

857

858

859

860

861

862

863

864

865

866

867

868

869

870

871

872

Fig. 2: Ca^{2+} influx induced by the PIEZO1 activator, YODA 1, in MCF-7 cells is dependent on cellular geometry.

(a) Representative fluorescence images of GCaMP6m-MCF-7 cells on “O” (left) and “L” (right) micropatterns. Scale bar = 50 μm . (b) Line scan (dotted line, scale bar = 20 μm) of YODA 1 (3 μM) induced changes in relative $[\text{Ca}^{2+}]_{\text{CYT}}$ (GCaMP6m fluorescence intensity in arbitrary units shown in

pseudocolour; scale bar = 200 s) over time in cells on “O” (top) and “L” micropatterns (bottom). (c) $[\text{Ca}^{2+}]_{\text{CYT}}$ changes in GCaMP6m-MCF-7 cells on “O” and “L” micropatterns with increasing YODA 1 concentrations. $[\text{Ca}^{2+}]_{\text{CYT}}$ changes in ten randomly selected representative cells on “O” (left) and “L” (middle) micropatterns and average of all cells (right). (d) $[\text{Ca}^{2+}]_{\text{CYT}}$ area under the curve (AUC), (e) sustained $[\text{Ca}^{2+}]_{\text{CYT}}$, last data point ($\Delta\text{F}/\text{F}_0$) and (f) maximum relative $[\text{Ca}^{2+}]_{\text{CYT}}$ change ($\Delta\text{F}/\text{F}_0$) for cells on “O” and “L” micropatterns induced by increasing concentrations of YODA 1 (0 – 3 μM). **From 0 – 750 s (top panel) and 750 s – end (bottom panel)**. Data shown represent the mean \pm S.E.M. (g) Top components from PCA captured differences between slow and fast decay. (h) Scatter plots of PC1 and PC2 scores of single cells activated by 3 μM YODA 1. (i) Means of “O” and “L” micropatterned cell clusters were significantly different at 3 μM YODA 1 (left). Linear discriminant analysis (LDA) shows that cluster means were best separated along a direction corresponding to the difference between fast and slow decay (right; i.e., positive LDA weight for PC1 and negative weight for PC2). (j) Reconstructed Ca^{2+} transients for cluster means show that “L” micropatterned cells decayed more slowly than “O” micropatterned cells at 3 μM YODA 1. (k) Illustration of a nonlinear model for Ca^{2+} transients incorporating an exponential rise, a slower exponential decay, and a steady-state constant (left). Representative model fits (median model error) for “O” and “L” micropatterned cells, respectively, at 3 μM YODA 1 (middle). Fitted decay time ($1/T_{\text{decay}}$) was significantly longer for “L”-shaped cells at 0.3, 1 and 3 μM YODA 1 (right; Y-axis is truncated to $1/T_{\text{decay}} = 1,000$ s) ($n = 4$ biological replicates). Unmarked $P \geq 0.05$, ** $P < 0.01$, *** $P < 0.001$, **** $P < 0.0001$.

884
 885
 886
 887
 888
 889
 890
 891
 892
 893
 894
 895
 896
 897
 898
 899
 900
 901

Fig. 4: The PIEZO1 activator, YODA 1, induces changes in the expression of specific EPI-MES markers in breast cancer cells.

(a) Densitometry and representative immunoblot showing downregulation of the epithelial marker E-cadherin in MDA-MB-468 by YODA 1 (72 h). (b) Densitometry and representative immunoblot showing YODA 1 induction of the mesenchymal marker, vimentin, in MDA-MB-468 cells (24 h). Data shown represent the mean ± S.E.M. (n = 4 biological replicates). (c) *CDH1*, *VIM*, *CLDN4* and *SERPINE1* mRNA levels in MDA-MB-468 cells stimulated with YODA 1 (3 h). Data shown represent the mean ± S.E.M. (n = 3 biological replicates). (d) Schematic representation of the strategy for *PIEZO1*-KO, gRNA sequences (g1, g2) in black, PAM sites in red. Arrows represent the positions of the designed primers for genomic PCR. (e) Knockdown of *PIEZO1* eliminates YODA 1 but not ATP-induced increases in [Ca²⁺]_{CYT} in MDA-MB-468 breast cancer cells; mean relative [Ca²⁺]_{CYT} (ΔF/F₀) from duplicate wells of three independent experiments. (f) Maximum relative [Ca²⁺]_{CYT} (ΔF/F₀) induced by YODA 1 (0 – 10 μM) (top) and 100 μM ATP (bottom). (g) *CDH1*, *VIM*, *CLDN4* and *SERPINE1* mRNA levels in MDA-MB-468-*PIEZO1*-KO cells and MDA-MB-468-scrambled cells stimulated with YODA 1 (3 h). Where appropriate, data shown represent the mean ± S.E.M. (n = 3 biological replicates). Unmarked P ≥ 0.05, * P < 0.05, ** P < 0.01, *** P < 0.001, **** P < 0.0001.

Supplementary Fig. 2

927
928
929
930
931
932
933
934
935
936
937
938
939

Supplementary Fig. 2: Principal components analysis and time series modelling further defines cell shape-dependent changes in YODA 1-induced Ca^{2+} influx.

(a) Principal component analysis (PCA) reveals that over 95% of the variance in the time series over both shapes and all concentrations can be explained by the top three components (PC1, PC2, PC3). Black curve shows the cumulative variance explained. Horizontal dotted line represents the percentage variance explained = 95%. (b) Fitting of nonlinear model for the quantification of Ca^{2+} decay time: data that meet the criteria for model fitting (left); data that do not meet the criteria for model fitting (right). The number of time series modelled are annotated as an n-value and percentage of the total sample. Refer to Supplementary Information for fitting criteria. (c) Model error for each cell at different YODA 1 concentrations. Scatter and violin plots for the sum of squared residuals (r^2 error) relative to the variance of the data for each shape and concentration. Unmarked $P \geq 0.05$, * $P < 0.05$, *** $P < 0.001$ ("O" vs "L").

Supplementary Fig. 4

949
 950
 951
 952
 953
 954
 955
 956
 957
 958
 959
 960

Supplementary Fig. 4: YODA 1-mediated Ca²⁺ influx induces changes in the expression of specific EPI-MES markers.

(a) YODA 1-induced [Ca²⁺]_{CYT} increases are through influx from extracellular Ca²⁺. Media free of extracellular Ca²⁺ (PSS BAPTA) eliminates YODA 1-induced increases in [Ca²⁺]_{CYT} in MDA-MB-468 breast cancer cells. (b) Maximum relative [Ca²⁺]_{CYT} (ΔF/F₀) induced by YODA 1 in media with (PSS Ca²⁺) and without extracellular Ca²⁺ (PSS BAPTA). (c) Assessment of mRNA expression of *CDH1*, *VIM*, *CLDN4* and *SERPINE1* in MDA-MB-468 cells in the presence of YODA 1 with and without BAPTA-AM (6 h). (d) Alterations in mRNA expression of *CDH1*, *VIM*, *CLDN4* and *SERPINE1* in MCF-7 cells induced by YODA 1 (24 h). Where appropriate, data shown represent the mean ± S.E.M. (n = 3 biological replicates). Unmarked P ≥ 0.05, * P < 0.05, ** P < 0.01, *** P < 0.001.

Supplementary Fig. 6

974
975 **Supplementary Fig. 6: CRISPR-Cas9-mediated *PIEZO1* knockout does not abolish EGF-mediated**
976 **changes in Ca^{2+} and gene expression in breast cancer cells.**

977 (a) Knockdown of *PIEZO1* does not eliminate epidermal growth factor (EGF)-induced increases in
978 $[Ca^{2+}]_{CYT}$ in MDA-MB-468 breast cancer cells. (b) $[Ca^{2+}]_{CYT}$ area under the curve (AUC) (left), sustained
979 $[Ca^{2+}]_{CYT}$, last data point ($\Delta F/F_0$) (middle) and maximum relative $[Ca^{2+}]_{CYT}$ change ($\Delta F/F_0$) (right) for cells
980 induced by different concentrations of EGF (0 – 50 ng/mL). (c) CRISPR-Cas9-mediated *PIEZO1* knockout
981 does not abolish EGF-mediated changes in gene expression in breast cancer cells. mRNA expression of
982 *CDH1*, *VIM*, *CLDN4* and *SERPINE1* in MDA-MB-468-*PIEZO1*-KO cells and MDA-MB-468-scrambled
983 cells stimulated by EGF (6 h). Data shown represent the mean \pm S.E.M. (n = 3 biological replicates).
984 Unmarked $P \geq 0.05$, * $P < 0.05$, ** $P < 0.01$.

5. The authors need to present a longitudinal plot indicating the short-term and longer-term calcium transients before and after YODA1 administration.

We have now made clearer (see above) that media addition controls were conducted (e.g. 0 μ M, Fig. 2c). We found that the MCF-7 cells did not demonstrate basal calcium transients in the absence of YODA1 (i.e., no spontaneous oscillations) for 30 min. After YODA1 administration, we followed the calcium transients for individual cells for 30 min (Fig. 2a) and found that most of the calcium responses triggered by YODA1 have recovered to basal levels by 30 min (see Fig. 2c) It is not possible with this throughput and temporal resolution to go beyond 30 min. The longer vs. shorter-term effects are addressed in our use of the Maximum relative $[Ca^{2+}]_{CYT}$ (peak is shorter term), area under the curve (AUC) and last data point (which considers long/sustained effects). We have added a new panel that addresses the reviewer's request to better understand the phenomena by adding Maximum relative $[Ca^{2+}]_{CYT}$ (0-750 s) and Maximum relative $[Ca^{2+}]_{CYT}$ (750 s-end) (Fig. 2f top and bottom panel) to make this comparison even clearer.

Fig. 2

848
849
850
851
852
853
854
855
856
857
858
859
860
861
862
863
864
865
866
867
868
869
870
871
872

Fig. 2: Ca^{2+} influx induced by the PIEZO1 activator, YODA 1, in MCF-7 cells is dependent on cellular geometry.

(a) Representative fluorescence images of GCaMP6m-MCF-7 cells on “O” (left) and “L” (right) micropatterns. Scale bar = 50 μm . (b) Line scan (dotted line, scale bar = 20 μm) of YODA 1 (3 μM) induced changes in relative $[\text{Ca}^{2+}]_{\text{CYT}}$ (GCaMP6m fluorescence intensity in arbitrary units shown in pseudocolour; scale bar = 200 s) over time in cells on “O” (top) and “L” micropatterns (bottom). (c) $[\text{Ca}^{2+}]_{\text{CYT}}$ changes in GCaMP6m-MCF-7 cells on “O” and “L” micropatterns with increasing YODA 1 concentrations. $[\text{Ca}^{2+}]_{\text{CYT}}$ changes in ten randomly selected representative cells on “O” (left) and “L” (middle) micropatterns and average of all cells (right). (d) $[\text{Ca}^{2+}]_{\text{CYT}}$ area under the curve (AUC), (e) sustained $[\text{Ca}^{2+}]_{\text{CYT}}$, last data point ($\Delta\text{F}/\text{F}_0$) and (f) maximum relative $[\text{Ca}^{2+}]_{\text{CYT}}$ change ($\Delta\text{F}/\text{F}_0$) for cells on “O” and “L” micropatterns induced by increasing concentrations of YODA 1 (0 – 3 μM). **From 0 – 750 s (top panel) and 750 s – end (bottom panel)**. Data shown represent the mean \pm S.E.M. (g) Top components from PCA captured differences between slow and fast decay. (h) Scatter plots of PC1 and PC2 scores of single cells activated by 3 μM YODA 1. (i) Means of “O” and “L” micropatterned cell clusters were significantly different at 3 μM YODA 1 (left). Linear discriminant analysis (LDA) shows that cluster means were best separated along a direction corresponding to the difference between fast and slow decay (right; i.e., positive LDA weight for PC1 and negative weight for PC2). (j) Reconstructed Ca^{2+} transients for cluster means show that “L” micropatterned cells decayed more slowly than “O” micropatterned cells at 3 μM YODA 1. (k) Illustration of a nonlinear model for Ca^{2+} transients incorporating an exponential rise, a slower exponential decay, and a steady-state constant (left). Representative model fits (median model error) for “O” and “L” micropatterned cells, respectively, at 3 μM YODA 1 (middle). Fitted decay time ($1/\tau_{\text{decay}}$) was significantly longer for “L”-shaped cells at 0.3, 1 and 3 μM YODA 1 (right; Y-axis is truncated to $1/\tau_{\text{decay}} = 1,000$ s) ($n = 4$ biological replicates). Unmarked $P \geq 0.05$, ** $P < 0.01$, *** $P < 0.001$, **** $P < 0.0001$.

6. It will be valuable to plot the frequency of calcium transients with and without yoda 1, rather than the amplitude. The steady state calcium transients are missing in these plots, which is a common phenotype in Gcamp6 based calcium imaging.

As noted above we have added arrows to make clear where we added media (control) or YODA1 in Fig. 2c. These single-celled MCF-7 cells (no cell-cell contacts) do not exhibit spontaneous calcium oscillations, some cells exhibited slow repeated increases in calcium after YODA1 stimulation, but these were very few (see Fig. 2c) and their impact would be reflected in our AUC data and Maximum relative $[Ca^{2+}]_{CYT}$ (0-750 s) and Maximum relative $[Ca^{2+}]_{CYT}$ (750 s-end) analyses (Fig. 2f top and bottom panel).

7. Coming back to the reaction kinetics of yoda1, it will be valuable to image for piezo1 channel using immunofluorescence. I am not convinced that administration of administration of Yoda1 will instantaneously increase the localization and activity of piezo1 channel. Also, doing so may help demonstrate if shape has a role in the localization of piezo1 on the plasma membrane.

We thank the reviewer for this suggestion. The ability of YODA1 to rapidly increase $[Ca^{2+}]_{CYT}$ in MCF-7 cells is shown in PIEZO1 CRISPR experiments (Fig. 4), which are definitive and also our various controls (liquid addition). We agree, that assessing PIEZO1 distribution in L vs. O would be interesting. But at the time when this part of the study was conducted in 2017, there were no satisfactory validated commercially available antibodies for IF or Western Blots. However, the reviewer's comment is insightful, and we have now included this (page 5 lines 223 – 224) as a suggestion for future studies.

“Other potential mechanisms could include ... or spatial distribution of PIEZO1 ...” (page 5 lines 223 – 224).

In conclusion, I think the authors need to revise the manuscript to address the mentioned concerns. This I believe will help strengthen the conclusions drawn by the authors.

We thank the reviewer for their suggestions and the opportunity to improve our manuscript for publication in Communications Biology.

Reviewer #2 (Remarks to the Author):

The manuscript by So et al identifies that changing cell morphology can modify Piezo1 expression and calcium signal that induces features of EPI-MES plasticity. The manuscript is interesting and suitable to the scope of Communications Biology. My main concerns are on the first part of the manuscript where the authors establish the correlation between cell geometry and Piezo1 expression. I also have some concern on the interpretation of the Ca signal and the Yap1 signal.

We thank the reviewer for their positive comments and appreciate the opportunity to clarify our work and improve our manuscript based on their feedback.

1. The main claims are based on two types of cell shapes (O and L) on 1100 μm^2 substrates. It's hard to generalize and claim that cell shape regulates Piezo1 expression, especially that measurements on 700 μm^2 do not seem to follow the same trend as on 1100 μm^2 substrates. More substrate sizes and shapes are needed to support the authors' claim.

We thank the reviewer for their suggestions. We have now included data assessing multiple shapes, which makes it clear why our focus on the elongated triangular morphology with associated changes to the text on page 2 lines 78 – 81, and the addition of Supplementary Fig. 1.

“Using commercially available surface micropatterning (refer to Materials and Methods), we then confined MCF-7 cells to various cellular geometry, and we found that the elongated triangular MCF-7 cells (“L” micropattern; 1,100 μm^2) demonstrated the most pronounced gene expression changes (Supplementary Fig. 1).” (page 2 lines 78 – 81)

Supplementary Fig. 1

919
 920 **Supplementary Fig. 1: 1,100 μm² “L” micropatterns changes the expression of EPI-MES and Ca²⁺**
 921 **signalling genes in MCF-7 epithelial like breast cancer cells.**
 922 Heatmap of expression changes (-ΔΔCt) including EPI-MES markers, Ca²⁺ channels and Ca²⁺ channel
 923 regulators and other genes on 1,100 μm² (left) or 700 μm² (right) micropatterned plates. Data represent
 924 the average gene expression changes normalized to unpatterned wells (-ΔΔCt) (n = 4 biological
 925 replicates, except TRPV1 n = 3 biological replicates). Unmarked P ≥ 0.05, * P < 0.05, ** P < 0.01, *** P <
 926 0.001 (“O” vs “L”).

2. does cell morphology alter local tensions in subcellular regions?

Indeed, Albert et al. have reported that changes in morphologies can alter local tensions in subcellular regions (PMID: 24896113). As the initial experiments (Fig. 1 and Supplementary Fig. 2) were conducted with the primary aim of accessing calcium signaling changes, we did not simultaneously assess local tensions in subcellular regions. However, we have now included in the discussion references to studies showing different subcellular tensions with different micropatterns (page 5, lines 225 – 229).

“As previous studies have found that micropatterning can create localized tension at specific subcellular regions³⁷, future studies could investigate PIEZO1 protein expression and the spatial distribution of PIEZO1 in cells adopting different cellular geometries and their correlation with local tensions at subcellular regions.” (page 5, lines 225 – 229)

3. what are the size of cells when they are not constrained to the patterned substrate? Are the cells stretched or compressed in the authors experiments?

We have now included the size of MCF-7 cells not constrained to the patterned substrate (page 5, lines 215 – 217), which is ~1017 μm² (from n = 448 segmented MCF-7 cells on an unpatterned surface). Hence, the 1,100 μm² pattern provided the most suitable and relevant surface area for our studies.

4. In figure 2, is the difference in Ca influx explained by different Piezo1 expression? If not, what are the molecular reasons that caused the different Ca dynamics? There is also an odd long-term increase of Ca signal on O cells.

We thank the reviewer for their comments. Indeed, we agree that the difference in Ca^{2+} influx could be explained not only by different PIEZO1 expression but also by factors such as the sensitivity of the PIEZO1 channel or other associated calcium signaling pathways. We have now added this to the discussion on page 5 lines 223 – 225, “The augmentation of intracellular Ca^{2+} increases induced by the PIEZO1 activator, YODA 1, in $1,100 \mu m^2$ “L” cells could be explained by the upregulation of PIEZO1 in these cells. Other potential mechanisms could include the sensitivity, kinetics, or spatial distribution of PIEZO1, or changes in other associated Ca^{2+} signaling pathways.”

We agree that the calcium signals seemed to increase slightly after the initial recovery from the peak in some cells and that this contributes to the sustained increase observed more often in the L shape. We have now expanded and significantly enhanced this section by adding a new panel in Fig. 2f. After YODA1 administration, we followed the calcium transients for individual cells for 30 min (Fig. 2a) and found that most of the calcium responses triggered by YODA1 had recovered to basal levels by 30 min (see Fig. 2c). It is not possible with this throughput and temporal resolution to go beyond 30 min. The longer vs. shorter-term effects are addressed in our use of the Maximum relative $[Ca^{2+}]_{CYT}$ (peak is shorter term), area under the curve (AUC) and last data point (which considers long/sustained effects). We have now added a new panel which we believe addresses the reviewer's request to better understand the phenomena by adding Maximum relative $[Ca^{2+}]_{CYT}$ (0-750 s) and Maximum relative $[Ca^{2+}]_{CYT}$ (750 s-end) (Fig. 2f top and bottom panel).

Fig. 2

848
849
850
851
852
853
854
855
856
857
858
859
860
861
862
863
864
865
866
867
868
869
870
871
872

Fig. 2: Ca^{2+} influx induced by the PIEZO1 activator, YODA 1, in MCF-7 cells is dependent on cellular geometry.

(a) Representative fluorescence images of GCaMP6m-MCF-7 cells on “O” (left) and “L” (right) micropatterns. Scale bar = 50 μm . (b) Line scan (dotted line, scale bar = 20 μm) of YODA 1 (3 μM) induced changes in relative $[\text{Ca}^{2+}]_{\text{CYT}}$ (GCaMP6m fluorescence intensity in arbitrary units shown in pseudocolour; scale bar = 200 s) over time in cells on “O” (top) and “L” micropatterns (bottom). (c) $[\text{Ca}^{2+}]_{\text{CYT}}$ changes in GCaMP6m-MCF-7 cells on “O” and “L” micropatterns with increasing YODA 1 concentrations. $[\text{Ca}^{2+}]_{\text{CYT}}$ changes in ten randomly selected representative cells on “O” (left) and “L” (middle) micropatterns and average of all cells (right). (d) $[\text{Ca}^{2+}]_{\text{CYT}}$ area under the curve (AUC), (e) sustained $[\text{Ca}^{2+}]_{\text{CYT}}$, last data point ($\Delta\text{F}/\text{F}_0$) and (f) maximum relative $[\text{Ca}^{2+}]_{\text{CYT}}$ change ($\Delta\text{F}/\text{F}_0$) for cells on “O” and “L” micropatterns induced by increasing concentrations of YODA 1 (0 – 3 μM). **From 0 – 750 s (top panel) and 750 s – end (bottom panel)**. Data shown represent the mean \pm S.E.M. (g) Top components from PCA captured differences between slow and fast decay. (h) Scatter plots of PC1 and PC2 scores of single cells activated by 3 μM YODA 1. (i) Means of “O” and “L” micropatterned cell clusters were significantly different at 3 μM YODA 1 (left). Linear discriminant analysis (LDA) shows that cluster means were best separated along a direction corresponding to the difference between fast and slow decay (right; i.e., positive LDA weight for PC1 and negative weight for PC2). (j) Reconstructed Ca^{2+} transients for cluster means show that “L” micropatterned cells decayed more slowly than “O” micropatterned cells at 3 μM YODA 1. (k) Illustration of a nonlinear model for Ca^{2+} transients incorporating an exponential rise, a slower exponential decay, and a steady-state constant (left). Representative model fits (median model error) for “O” and “L” micropatterned cells, respectively, at 3 μM YODA 1 (middle). Fitted decay time ($1/\tau_{\text{decay}}$) was significantly longer for “L”-shaped cells at 0.3, 1 and 3 μM YODA 1 (right; Y-axis is truncated to $1/\tau_{\text{decay}} = 1,000$ s) ($n = 4$ biological replicates). Unmarked $P \geq 0.05$, ** $P < 0.01$, *** $P < 0.001$, **** $P < 0.0001$.

5. Is the Piezo1-induced changes in marker expression solely a Ca effect? Can these changes be induced by increasing intracellular Ca²⁺ level while not activating Piezo1?

Calcium signaling indeed affects changes in marker expression (PMID: 23686305, PMID: 33322037), and this is now discussed on page 5 lines 244 – 245 “We and others have shown that Ca²⁺ signaling influences EPI-MES plasticity in breast cancer cells^{29,42”}. In our studies, we show that the changes in marker expression are related to PIEZO1 induced-Ca²⁺ increases as YODA 1-induced activation of PIEZO1 resulted in changes in EMT markers such as SERPINE1. We now provide further evidence for this link via new experiments showing that SERPINE1 levels are very sensitive to calcium and that chelating intracellular Ca²⁺ using BAPTA-AM eliminates SERPINE1 induction by YODA 1 (Supplementary Fig. 4c). This work is also described in page 4 lines 170 – 173 of the new manuscript “SERPINE1 levels were particularly sensitive to intracellular Ca²⁺ levels and its upregulation by YODA 1 was Ca²⁺ signal-dependent as demonstrated by the effects of intracellular Ca²⁺ chelation with BAPTA-AM (Supplementary Fig. 4c)”.

Supplementary Fig. 4

949
 950
 951
 952
 953
 954
 955
 956
 957
 958
 959
 960

Supplementary Fig. 4: YODA 1-mediated Ca²⁺ influx induces changes in the expression of specific EPI-MES markers.

(a) YODA 1-induced [Ca²⁺]_{CYT} increases are through influx from extracellular Ca²⁺. Media free of extracellular Ca²⁺ (PSS BAPTA) eliminates YODA 1-induced increases in [Ca²⁺]_{CYT} in MDA-MB-468 breast cancer cells. (b) Maximum relative [Ca²⁺]_{CYT} (ΔF/F₀) induced by YODA 1 in media with (PSS Ca²⁺) and without extracellular Ca²⁺ (PSS BAPTA). (c) Assessment of mRNA expression of *CDH1*, *VIM*, *CLDN4* and *SERPINE1* in MDA-MB-468 cells in the presence of YODA 1 with and without BAPTA-AM (6 h). (d) Alterations in mRNA expression of *CDH1*, *VIM*, *CLDN4* and *SERPINE1* in MCF-7 cells induced by YODA 1 (24 h). Where appropriate, data shown represent the mean ± S.E.M. (n = 3 biological replicates). Unmarked P ≥ 0.05, * P < 0.05, ** P < 0.01, *** P < 0.001.

6. In figure 5a, the changes in Yap1 level is very unclear.

Franklin et al. (PMID: 32917893) recently characterized that YAP1 activity does not have a simple linear correlation with its nuclear localization but is dependent on specific nuclear-cytoplasmic exchange. They characterized a localization-reset phenomenon that precedes YAP1 target gene expression. They also found that calcium signaling can trigger the localization-reset of YAP1. In agreement with their findings, we observed the same "initial depletion and subsequent accumulation of nuclear YAP1" (page 4 line 193) in MDA-MB-468 cells after YODA1 (3 μ M) activation. We measured the YAP1 localization-reset at single-cell (Fig. 5b, top and middle panel) and the population level (Fig. 5b bottom panel). We observed the reset at the second time point of YODA1 addition, shown as a dip in nuclear YAP1 in the YODA1 treated cells.

We have now added in the manuscript "Here, we also observed a decrease in nuclear YAP1 upon YODA1 treatment at 15 min and a subsequent "reset" at 1 h time point (Fig. 5a and b)." (page 4 lines 194 – 196)

We have also used dotted lines to indicate the nucleus in Fig. 5a (left panel) to make clearer the nuclear and cytoplasmic regions.

Fig. 5

904 **Fig. 5: YODA 1 induces localization-reset of YAP1 and upregulates YAP1 target gene SERPINE1 in**
 905 **breast cancer cells.**

906 (a) Representative images showing immunofluorescence of YAP1 with dotted lines indicating the nuclei
 907 (left) and nuclei (DAPI, middle) of MDA-MB-468 cells at 0 min (basal), 15 min, 1 h and 3 h following 0 μM
 908 and 3 μM YODA 1 stimulation. Scale bar = 20 μm. Zoomed-in images showing YAP1 localization and the
 909 nucleus of a representative cell (right, scale bar = 10 μm). (b) YODA 1 induced an initial depletion (15
 910 min) and a subsequent accumulation of nuclear YAP1. Nuclear YAP1 changes quantitated as correlation
 911 coefficient, represented as violin plots (top), histograms (middle) and means (bottom). (c) mRNA
 912 expression of *CTGF* and *SERPINE1* in MDA-MB-468 cells after ON-TARGETplus SMARTpool siRNA-
 913 mediated silencing of *YAP1*, *TEAD4* and *WWTR1* (encodes for TAZ). Data shown represent the mean ±
 914 S.E.M. (n = 3 biological replicates). ns P ≥ 0.05, * P < 0.05, ** P < 0.01, *** P < 0.001, **** P < 0.0001.

Reviewer #3 (Remarks to the Author):

In this manuscript, the authors explore the potential role of the mechanosensitive ion channel PIEZO1 in cell shape and plasticity. In addition to investigating the link between Ca²⁺ signalling, cell shape and epithelial to mesenchymal plasticity. Of particular interest is the link between PIEZO1 and SERPINE1 expression and other genes involved in epithelial to mesenchymal plasticity. The role of PIEZO1 in cell morphology is an emerging area and the work presented is novel and of interest to the field. The manuscript overall is very well written and presented.

We thank the reviewer for their positive comments.

Major comments

1. Are protein levels of Piezo1 changed in o versus L shaped cells? please provide western blots of protein expression levels for MCF 7 cells cultured in O or L conformations, if technically feasible.

We thank the reviewer for the comments and for understanding that harvesting proteins from micropatterned cells is technically challenging. Indeed, the yield of protein from each well would be too low, as each well has only ~3000 cells. In addition, at the time when this part of the study was conducted in 2017, there were no appropriate commercially available antibodies for IF or Western Blots. However, the reviewer's comment is definitely valid, and we have now included such studies in our discussion for future work (page 5, lines 227 – 228) "future studies could investigate PIEZO1 protein expression ... in cells adopting different cellular geometries..."

2. Are the channel kinetics affected by changes in cell shape or is the increased calcium influx due to increased protein expression? Please provide evidence using patch clamp.

Patching these cells on these plates at a sufficient number is beyond the scope of this study. However, we now discuss this possible future work in the manuscript (page 5, lines 229 – 231) "Future work could also evaluate the relative contributions of changes in PIEZO1 levels and/or channel kinetics to the shape-dependent increases in calcium influx induced by YODA 1 that have been observed in our studies".

3. Does PIEZO1 knockdown abrogate changes in cell morphology in MCF7 cell cultured in O and L morphologies? The group have used the necessary CRISPR/Cas9 PIEZO1 knockout technology or transient knockout capabilities in this manuscript.

Assessing the ability of PIEZO1 KO to influence the ability of cells to adopt different morphologies (e.g., L vs. O) is not straightforward and was not the aim of this study. It would be hard to determine how a change in % of cells attaching to L patterns might reflect an ability to adopt a shape change vs. a change in attachment efficiency. However, we now cite papers that do report an association with PIEZO1 and cell morphology, as this possible role for PIEZO1 raised by the reviewer is important.

"PIEZO1 is associated with cellular organization and upregulation of PIEZO1 occurs in disorganized malignant T4-2 breast cells compared with non-malignant S1 cells that grow in organized acini³⁸" (page 5 lines 233 – 235)

"Our results show that in addition to being a ... regulator of cellular morphology in CHO- α 4WT cells⁷ and muscle stem cells⁴¹, PIEZO1 can be remodeled as a consequence of changes in cell shape ..." (page 5 lines 240 – 241)

4. Is PIEZO1 sensitivity to mechanical stimulation (as opposed to pharmacological stimulation) affected during changes in cell morphology in the O and L morphologies?

We thank the reviewer for this comment. We, like the reviewer, were also curious if PIEZO1 sensitivity to mechanical stimulation (as opposed to pharmacological stimulation) was affected during changes in cell morphology in the O and L morphologies. However, we did not observe that the L cells were more mechanically sensitive. During compound addition using our high-throughput imaging system, we sometimes noticed a "touch response" in mechanically sensitive cell lines, during which the addition of

liquid (without pharmacological activators) could induce a calcium response (see minor change in Fig. 2c, O chape 0 μM YODA (Media)). However, we did not observe a difference in “touch” sensitivity during the addition of DMSO control in O and L cells and this is reflected in no difference in panels d-f in Fig 2. It would not be possible to use a commercial or custom-built device to dynamically alter cell geometry in a high-throughput calcium imaging device. Such an advance would be a transformative breakthrough in the field and is therefore beyond the scope of this paper.

5. Please confirm changes in PIEZO1 protein expression in MDA-MB-468 cells as well as in knockdowns.

When these experiments were conducted, there were no commercially available, CRISPR-validated PIEZO1 antibodies, and hence we relied on functional calcium influx assays (Fig. 4e). However, as we had retained the samples for this paper, we conducted new experiments with a now commercially available antibody, and we demonstrated PIEZO1 knockdown at the protein level using immunoblotting (Supplementary Fig. 5a). Our genotyping (Methods and Supplementary Table 1) also showed a high knockdown score of ~80 and an indel percentage of ~90%.

We have now added to the manuscript a new figure (Supplementary Fig. 5a), showing the results and relevant information in Materials and Methods “The antibodies used were anti-PIEZO1 antibody (1:1000 in 5% w/v milk PBST, NBP275617, Novus Biologicals)... (Page 9)”

Supplementary Fig. 5

962
963
964
965
966
967
968
969
970
971
972

Supplementary Fig. 5: The effect of CRISPR-Cas9-mediated PIEZO1 knockout on basal EPI-MES marker expression and YODA 1-mediated protein expression changes.

(a) Densitometry and representative immunoblot showing knockdown of PIEZO1 in MDA-MB-468 cells. **(b)** CRISPR-Cas9-mediated PIEZO1 knockout does not change the basal mRNA expression of CDH1, VIM, CLDN4 and SERPINE1 in MDA-MB-468 cells. **(c)** Densitometry and representative immunoblot showing knockdown of PIEZO1 eliminates the downregulation of the epithelial marker, E-cadherin, in MDA-MB-468 by YODA 1 (72 h). **(d)** Densitometry and representative immunoblot showing knockdown of PIEZO1 changes YODA 1-mediated induction of the mesenchymal marker vimentin in MDA-MB-468 cells (24 h). Data shown represent the mean \pm S.E.M. (n = 3 biological replicates). ns P \geq 0.05, * P < 0.05, ** P < 0.01.

Minor Comments

6. The work proposes that the increase in PIEZO1 activity is due to increased gene expression over the 16-24 hour culture period. How does this culture time scale compare to physiological relevant changes in cell shape? Please provide more description of the physiological relevance of the O/L pattern culture system.

We thank the reviewer for the comments. During cancer metastasis, cancer cells change their shape as they squeeze through blood vessels. When quantifying in vitro, such as using scratch assay, the migration of cancer cells can occur in hours (PMID: 17406593, PMID: 15576902). A 16–24-hour culture period is sufficient to recapitulate the shape changes that occur during cancer cell migration. Here, we confined breast cancer cells to the non-polarized rounded ("O" micropatterned) pattern, typical of non-migrating cells, and "L", which created a single free-edge organization.

We have now included "Fig. 1b-d shows the comparison between non-polarized rounded ("O" micropatterned) and elongated triangular ("L" micropatterned) geometries (Fig. 1b-d)." (page 2 lines 81 – 82), and "The elongated triangular cellular geometry creates a long single free-edge organization not seen in other micropatterns used in this study." (page 5 lines 217 – 219)

7. Nonomura et al, Friedrich et al and Chuntharpursat-Bon et al, have previously reported changes in VE cadherin/B-actin upon YODA1 treatment in endothelial cells. The significance e-cadherin should be discussed in more detail and how it relates to physiological epithelial remodelling.

We thank the reviewer for referring us to the relevant studies. We have now included this work in our updated discussion (page 5 lines 254 – 259) "One feature of EPI-MES plasticity is the loss or decrease of junctional protein E-cadherin¹. In MDA-MB-468 breast cancer cells, YODA 1 activation resulted in an increase in vimentin and a decrease in E-cadherin. The association of PIEZO1 and junctional proteins is also seen through the interaction of Piezo1 with PECAM1⁴⁹, and Piezo1-induced remodeling of junctional proteins in endothelial cells and lymphatic endothelial cells^{50,51}. Future studies could further evaluate the intersection of PIEZO1 and junctional proteins in the context of cancer progression pathways."

8. The interpretation of the results should be in a separate discussion section and should go into more detail for each conclusion.

We thank the reviewer for the comments. We have now included a separate discussion section and have expanded our discussion as suggested (pages 4 – 6).

REVIEWERS' COMMENTS:

Reviewer #1 (Remarks to the Author):

I thank the authors for providing explanation/answers to my comments in sufficient detail. I am not convinced by the rebuttal to comment#3 that a cell stretching device is not possible to build or buy off the shelf. But the authors have astutely removed the word 'stretch' from the text to present the data in a different context.

However, given the scope of the paper and the journal, I will concede that the new experiments and modified figures and associated texts have improved the manuscript significantly. Therefore, I recommend this manuscript for publication.

Reviewer #3 (Remarks to the Author):

The authors have addressed all concerns. Where technically possible they have included new data and otherwise provided further detail in the text. They have expanded the experimental detail and interpretation of their data. I would like to thank the authors for an elegant manuscript and their effort in addressing the extensive review comments.

Reviewers' comments:

Reviewer #1 (Remarks to the Author):

Summary:

Here the authors have demonstrated that changes in geometric pattern (shape) from circle to a triangle can trigger upregulation of mesenchymal genes due to increased calcium signaling in a Piezo-1 dependent manner. Using genetic and drug perturbations, the authors have further showed that Piezo-1 mediated calcium signaling can alter Yap localization and upregulation of EPI-MES markers.

Comments:

I think this is an interesting observation and I want to commend the authors for conducting a detailed investigation into role of calcium signaling in epi-mes transition. I enjoyed reading the manuscript and learned a few things along the way. I think the authors have done robust statistical analysis of calcium signaling and correlated it with gene expression. Especially, I find the correlation between shape and epi-mes transition fascinating and intriguing.

However, I have certain misgivings regarding the experimental methods and interpretation of the results. I think the authors need to address this to improve the quality of the manuscript and its conclusions.

We thank the reviewer for their positive and encouraging comments and are glad they enjoyed reading our manuscript. We are excited to further improve the paper based on this reviewer's suggestions.

1. It is unclear to me why the authors arbitrarily decide on a circular and L shaped pattern. Maybe I missed something here, but it would be good to give an explanation to the readers about the rationale of the geometric patterns. For instance, if the objective was to compare elongated vs rounded shaped cells, the authors could have designed a rectangular pattern.

We thank the reviewer for raising this point. We conducted initial qRT-PCR studies in many shapes (CYTOOplates 96 Starter-A, commercially available), and we have now included this early data in the manuscript to provide more context (Supplementary Fig. 1), which showed that gene expression changes were more pronounced in 1100 μm^2 L, which was the reason why we focused on this shape for further study. L is known to create a single free-edge organization (see figure below from the CYTOO website <https://cytoo.com/micropattern-products/plates/cytooplates%E2%84%A2-96-rw-starter>), showing L represents a single free edge organization). This detail has also now been added to the manuscript (page 2 lines 78 – 82 and Supplementary Fig. 1).

A whole range of micropatterns for diverse applications

	Disc	Crossbow	H	Y	L
Micropatterns					Cells					Description	No polarization	Strong polarization	Symmetric organization	Triaxial symmetry organization	Single free edge organization
Noteworthy Applications	Cell arraying Ciliogenesis	Cell polarity Organelle positioning Receptor internalization	Cell division Cell-cell junction	Multipolar division	Cytoskeleton rearrangement & Spindle orientation

Supplementary Fig. 1

919
920
921
922
923
924
925
926

Supplementary Fig. 1: 1,100 μm^2 "L" micropatterns changes the expression of EPI-MES and Ca^{2+} signalling genes in MCF-7 epithelial like breast cancer cells.

Heatmap of expression changes ($-\Delta\Delta C_t$) including EPI-MES markers, Ca^{2+} channels and Ca^{2+} channel regulators and other genes on 1,100 μm^2 (left) or 700 μm^2 (right) micropatterned plates. Data represent the average gene expression changes normalized to unpatterned wells ($-\Delta\Delta C_t$) (n = 4 biological replicates, except *TRPV1* n = 3 biological replicates). Unmarked $P \geq 0.05$, * $P < 0.05$, ** $P < 0.01$, *** $P < 0.001$ ("O" vs "L").

"Using commercially available surface micropatterning (refer to Materials and Methods), we then confined MCF-7 cells to various cellular geometry, and we found that the elongated triangular MCF-7 cells ("L" micropattern; 1,100 μm^2) demonstrated the most pronounced gene expression changes (Supplementary Fig. 1). Fig. 1b-d shows the comparison between non-polarized rounded ("O" micropatterned) and elongated triangular ("L" micropatterned) geometries (Fig. 1b-d)." (page 2 lines 78 – 82)

2. Changes in geometric pattern from O to L is not equivalent to membrane stretching as the surface area is still the same. Also why did the authors chose only two areas- 700 and 1100 μm^2 ? Nothing interesting happens in both O and L-shaped patterns with area of 700 μm^2 , the authors should do similar experiment a pattern larger than 1100 μm^2 . This will indicate if there is anything special about 1100 μm^2 or the geometry plays a significant role only in stretched out cells. It is interesting why in smaller area there is no difference in gene expression between O and L shaped patterns.

In terms of 700 and 1100 μm^2 , we now note in the manuscript that the 1100 μm^2 is closer to the surface area of MCF-7 cells where the shape is not confined see page 5 lines 215 – 217 (~1017 μm^2 , from 448

segmented MCF-7 cells). The larger surface area of micropatterns (1,600 μm^2) would not be biologically relevant for MCF-7 cells. We chose 700 and 1100 μm^2 due to our use of high-content imaging equipment; we were limited to the use of commercially available cell surface pattern microplates (CYTOOplates 96 Starter-A) – the larger surface area micropatterns from the company are only available as one full 96-well plate for one micropattern (1600 μm^2) – this posed a limitation for our initial aim of screening for “interesting” shapes.

We agree with the reviewer that the term “stretching” for L vs. O at 1100 μm^2 lacks sufficient clarity and could cause confusion, so we no longer use this term. We agree with the reviewer (as reflected in our original title), that it is shape / cellular geometry that is the feature which drives our reported changes. In this context, we have added as a supplementary figure data from other shapes showing that the gene expression changes are unique to “L”. This is now discussed in detail in lines 212 – 215 “The most pronounced gene expression changes were observed in the 1,100 μm^2 elongated triangular but not at 700 μm^2 , when compared to other shapes. The maximum distance between two points was 25% greater for 1,100 μm^2 surfaces than for 700 μm^2 micropattern surfaces: 37.4 μm compared to 29.9 μm (“O” micropattern) and 66.3 μm compared to 52.9 μm (“L” micropattern).” and shown in Supplementary Fig. 1. The L micropattern is known to create a single free-edge organization; hence we have changed from using the term “stretched” to elongated triangular geometry or single free-edge organization.

3. To claim that indeed the increased calcium spikes are due to stretching and not geometric patterns, the authors should perform a cell stretching experiment. This will help decouple cellular stretch from geometric configuration, which might be a perturbation on its own. To do this experiment, I suggest the authors can use a custom built or off-the-shelf cell stretcher.

As discussed above we have now removed the term stretching from the paper. In a high throughput calcium imaging instrument that was required to fully define the calcium signaling changes observed in our work, it would not be possible to use a custom-built or off-the-shelf cell stretcher to dynamically alter cell geometry. Such an advancement would be a transformative breakthrough in the field and is beyond the scope of this paper.

4. The experiments with YODA1 are interesting. However, it is not clear what the reaction kinetics of yoda1 in these cells is. It's unclear in these cells, when was YODA1 added and how much time does it take for it to change cellular behavior. Cells can have calcium spikes even when media is changed. Here, the authors need to do a control with changing media without yoda1 as a control. Any mechanical perturbation including changing media will cause a calcium spike.

We thank the reviewer for their comments and we are happy to clarify and add further details regarding our Ca^{2+} measurement protocols. In all the calcium experiments (Figs. 2, 4, and Supplementary Figs. 2, 4, 6), YODA1 was added using a custom-created protocol on our automated high-content microscope (IXM) or automatically by our high-throughput system (FLIPR). In brief, cells were plated in 96-well sample plates (100 μL media), and DMSO control or YODA1 (six times or three times the final desired concentrations for IXM and FLIPR, respectively) was diluted using media in separate 96-well compound plates. The drugs were aspirated from the compound plates (20 μL and 50 μL for IXM and FLIPR, respectively), added to the sample plate, and the calcium signaling changes were continuously recorded. Therefore, the addition of DMSO (control) also represented the control for any mechanical perturbation as a consequence of liquid addition.

We have now added arrows (below the x-axis) on all our figures to show where control (media addition) and/or YODA or ATP were added, and we have changed the text on pages 8 and 9 (Materials and Methods) to make this clearer.

“YODA 1 solutions were prepared in FluoroBrite DMEM, supplemented with 0.5% v/v FBS, 4 mM L-glutamine, and 25 mM HEPES buffer to obtain the final concentrations as stated. At the tenth time point, 20 μL of the prepared YODA 1 was added to the micropatterned plates. Images were continually acquired before, during and after YODA 1 addition.” (page 8 lines 386 – 390)

“Unless otherwise stated, YODA 1, ATP and hEGF were prepared on a compound plate. FLIPR^{TETRA} was

programmed to automatically add 50 μL to the cell-containing sample plates to obtain the final concentrations as stated in figures and Ca^{2+} signaling changes were continuously assessed before, during and after reagent addition.” (page 9 lines 462 – 465)

Fig. 2

848

849

850

851

852

853

854

855

856

857

858

859

860

861

862

863

864

865

866

867

868

869

870

871

872

Fig. 2: Ca^{2+} influx induced by the PIEZO1 activator, YODA 1, in MCF-7 cells is dependent on cellular geometry.

(a) Representative fluorescence images of GCaMP6m-MCF-7 cells on “O” (left) and “L” (right) micropatterns. Scale bar = 50 μm . (b) Line scan (dotted line, scale bar = 20 μm) of YODA 1 (3 μM) induced changes in relative $[\text{Ca}^{2+}]_{\text{CYT}}$ (GCaMP6m fluorescence intensity in arbitrary units shown in

pseudocolour; scale bar = 200 s) over time in cells on “O” (top) and “L” micropatterns (bottom). (c) $[\text{Ca}^{2+}]_{\text{CYT}}$ changes in GCaMP6m-MCF-7 cells on “O” and “L” micropatterns with increasing YODA 1 concentrations. $[\text{Ca}^{2+}]_{\text{CYT}}$ changes in ten randomly selected representative cells on “O” (left) and “L” micropatterns and average of all cells (right). (d) $[\text{Ca}^{2+}]_{\text{CYT}}$ area under the curve (AUC), (e) sustained $[\text{Ca}^{2+}]_{\text{CYT}}$, last data point ($\Delta\text{F}/\text{F}_0$) and (f) maximum relative $[\text{Ca}^{2+}]_{\text{CYT}}$ change ($\Delta\text{F}/\text{F}_0$) for cells on “O” and “L” micropatterns induced by increasing concentrations of YODA 1 (0 – 3 μM). **From 0 – 750 s (top panel) and 750 s – end (bottom panel)**. Data shown represent the mean \pm S.E.M. (g) Top components from PCA captured differences between slow and fast decay. (h) Scatter plots of PC1 and PC2 scores of single cells activated by 3 μM YODA 1. (i) Means of “O” and “L” micropatterned cell clusters were significantly different at 3 μM YODA 1 (left). Linear discriminant analysis (LDA) shows that cluster means were best separated along a direction corresponding to the difference between fast and slow decay (right; i.e., positive LDA weight for PC1 and negative weight for PC2). (j) Reconstructed Ca^{2+} transients for cluster means show that “L” micropatterned cells decayed more slowly than “O” micropatterned cells at 3 μM YODA 1. (k) Illustration of a nonlinear model for Ca^{2+} transients incorporating an exponential rise, a slower exponential decay, and a steady-state constant (left). Representative model fits (median model error) for “O” and “L” micropatterned cells, respectively, at 3 μM YODA 1 (middle). Fitted decay time ($1/T_{\text{decay}}$) was significantly longer for “L”-shaped cells at 0.3, 1 and 3 μM YODA 1 (right; Y-axis is truncated to $1/T_{\text{decay}} = 1,000$ s) ($n = 4$ biological replicates). Unmarked $P \geq 0.05$, ** $P < 0.01$, *** $P < 0.001$, **** $P < 0.0001$.

884
 885
 886
 887
 888
 889
 890
 891
 892
 893
 894
 895
 896
 897
 898
 899
 900
 901

Fig. 4: The PIEZO1 activator, YODA 1, induces changes in the expression of specific EPI-MES markers in breast cancer cells.

(a) Densitometry and representative immunoblot showing downregulation of the epithelial marker E-cadherin in MDA-MB-468 by YODA 1 (72 h). (b) Densitometry and representative immunoblot showing YODA 1 induction of the mesenchymal marker, vimentin, in MDA-MB-468 cells (24 h). Data shown represent the mean ± S.E.M. (n = 4 biological replicates). (c) *CDH1*, *VIM*, *CLDN4* and *SERPINE1* mRNA levels in MDA-MB-468 cells stimulated with YODA 1 (3 h). Data shown represent the mean ± S.E.M. (n = 3 biological replicates). (d) Schematic representation of the strategy for *PIEZO1*-KO, gRNA sequences (g1, g2) in black, PAM sites in red. Arrows represent the positions of the designed primers for genomic PCR. (e) Knockdown of *PIEZO1* eliminates YODA 1 but not ATP-induced increases in [Ca²⁺]_{CYT} in MDA-MB-468 breast cancer cells; mean relative [Ca²⁺]_{CYT} (ΔF/F₀) from duplicate wells of three independent experiments. (f) Maximum relative [Ca²⁺]_{CYT} (ΔF/F₀) induced by YODA 1 (0 – 10 μM) (top) and 100 μM ATP (bottom). (g) *CDH1*, *VIM*, *CLDN4* and *SERPINE1* mRNA levels in MDA-MB-468-*PIEZO1*-KO cells and MDA-MB-468-scrambled cells stimulated with YODA 1 (3 h). Where appropriate, data shown represent the mean ± S.E.M. (n = 3 biological replicates). Unmarked P ≥ 0.05, * P < 0.05, ** P < 0.01, *** P < 0.001, **** P < 0.0001.

Supplementary Fig. 2

927
928
929
930
931
932
933
934
935
936
937
938
939

Supplementary Fig. 2: Principal components analysis and time series modelling further defines cell shape-dependent changes in YODA 1-induced Ca^{2+} influx.

(a) Principal component analysis (PCA) reveals that over 95% of the variance in the time series over both shapes and all concentrations can be explained by the top three components (PC1, PC2, PC3). Black curve shows the cumulative variance explained. Horizontal dotted line represents the percentage variance explained = 95%. (b) Fitting of nonlinear model for the quantification of Ca^{2+} decay time: data that meet the criteria for model fitting (left); data that do not meet the criteria for model fitting (right). The number of time series modelled are annotated as an n-value and percentage of the total sample. Refer to Supplementary Information for fitting criteria. (c) Model error for each cell at different YODA 1 concentrations. Scatter and violin plots for the sum of squared residuals (r^2 error) relative to the variance of the data for each shape and concentration. Unmarked $P \geq 0.05$, * $P < 0.05$, *** $P < 0.001$ ("O" vs "L").

Supplementary Fig. 4

949
950
951
952
953
954
955
956
957
958
959
960

Supplementary Fig. 4: YODA 1-mediated Ca²⁺ influx induces changes in the expression of specific EPI-MES markers.

(a) YODA 1-induced [Ca²⁺]_{CYT} increases are through influx from extracellular Ca²⁺. Media free of extracellular Ca²⁺ (PSS BAPTA) eliminates YODA 1-induced increases in [Ca²⁺]_{CYT} in MDA-MB-468 breast cancer cells. (b) Maximum relative [Ca²⁺]_{CYT} (ΔF/F₀) induced by YODA 1 in media with (PSS Ca²⁺) and without extracellular Ca²⁺ (PSS BAPTA). (c) Assessment of mRNA expression of *CDH1*, *VIM*, *CLDN4* and *SERPINE1* in MDA-MB-468 cells in the presence of YODA 1 with and without BAPTA-AM (6 h). (d) Alterations in mRNA expression of *CDH1*, *VIM*, *CLDN4* and *SERPINE1* in MCF-7 cells induced by YODA 1 (24 h). Where appropriate, data shown represent the mean ± S.E.M. (n = 3 biological replicates). Unmarked P ≥ 0.05, * P < 0.05, ** P < 0.01, *** P < 0.001.

Supplementary Fig. 6

974
 975 **Supplementary Fig. 6: CRISPR-Cas9-mediated *PIEZO1* knockout does not abolish EGF-mediated**
 976 **changes in Ca^{2+} and gene expression in breast cancer cells.**

977 (a) Knockdown of *PIEZO1* does not eliminate epidermal growth factor (EGF)-induced increases in
 978 $[Ca^{2+}]_{CYT}$ in MDA-MB-468 breast cancer cells. (b) $[Ca^{2+}]_{CYT}$ area under the curve (AUC) (left), sustained
 979 $[Ca^{2+}]_{CYT}$, last data point ($\Delta F/F_0$) (middle) and maximum relative $[Ca^{2+}]_{CYT}$ change ($\Delta F/F_0$) (right) for cells
 980 induced by different concentrations of EGF (0 – 50 ng/mL). (c) CRISPR-Cas9-mediated *PIEZO1* knockout
 981 does not abolish EGF-mediated changes in gene expression in breast cancer cells. mRNA expression of
 982 *CDH1*, *VIM*, *CLDN4* and *SERPINE1* in MDA-MB-468-*PIEZO1*-KO cells and MDA-MB-468-scrambled
 983 cells stimulated by EGF (6 h). Data shown represent the mean \pm S.E.M. (n = 3 biological replicates).
 984 Unmarked $P \geq 0.05$, * $P < 0.05$, ** $P < 0.01$.

5. The authors need to present a longitudinal plot indicating the short-term and longer- term calcium transients before and after YODA1 administration.

We have now made clearer (see above) that media addition controls were conducted (e.g. 0 μ M, Fig. 2c). We found that the MCF-7 cells did not demonstrate basal calcium transients in the absence of YODA1 (i.e., no spontaneous oscillations) for 30 min. After YODA1 administration, we followed the calcium transients for individual cells for 30 min (Fig. 2a) and found that most of the calcium responses triggered by YODA1 have recovered to basal levels by 30 min (see Fig. 2c) It is not possible with this throughput and temporal resolution to go beyond 30 min. The longer vs. shorter-term effects are addressed in our use of the Maximum relative $[Ca^{2+}]_{CYT}$ (peak is shorter term), area under the curve (AUC) and last data point (which considers long/sustained effects). We have added a new panel that addresses the reviewer's request to better understand the phenomena by adding Maximum relative $[Ca^{2+}]_{CYT}$ (0-750 s) and Maximum relative $[Ca^{2+}]_{CYT}$ (750 s-end) (Fig. 2f top and bottom panel) to make this comparison even clearer.

Fig. 2

848
849
850
851
852
853
854
855
856
857
858
859
860
861
862
863
864
865
866
867
868
869
870
871
872

Fig. 2: Ca^{2+} influx induced by the PIEZO1 activator, YODA 1, in MCF-7 cells is dependent on cellular geometry.

(a) Representative fluorescence images of GCaMP6m-MCF-7 cells on “O” (left) and “L” (right) micropatterns. Scale bar = 50 μm . (b) Line scan (dotted line, scale bar = 20 μm) of YODA 1 (3 μM) induced changes in relative $[\text{Ca}^{2+}]_{\text{CYT}}$ (GCaMP6m fluorescence intensity in arbitrary units shown in pseudocolour; scale bar = 200 s) over time in cells on “O” (top) and “L” micropatterns (bottom). (c) $[\text{Ca}^{2+}]_{\text{CYT}}$ changes in GCaMP6m-MCF-7 cells on “O” and “L” micropatterns with increasing YODA 1 concentrations. $[\text{Ca}^{2+}]_{\text{CYT}}$ changes in ten randomly selected representative cells on “O” (left) and “L” (middle) micropatterns and average of all cells (right). (d) $[\text{Ca}^{2+}]_{\text{CYT}}$ area under the curve (AUC), (e) sustained $[\text{Ca}^{2+}]_{\text{CYT}}$, last data point ($\Delta\text{F}/\text{F}_0$) and (f) maximum relative $[\text{Ca}^{2+}]_{\text{CYT}}$ change ($\Delta\text{F}/\text{F}_0$) for cells on “O” and “L” micropatterns induced by increasing concentrations of YODA 1 (0 – 3 μM). **From 0 – 750 s (top panel) and 750 s – end (bottom panel)**. Data shown represent the mean \pm S.E.M. (g) Top components from PCA captured differences between slow and fast decay. (h) Scatter plots of PC1 and PC2 scores of single cells activated by 3 μM YODA 1. (i) Means of “O” and “L” micropatterned cell clusters were significantly different at 3 μM YODA 1 (left). Linear discriminant analysis (LDA) shows that cluster means were best separated along a direction corresponding to the difference between fast and slow decay (right; i.e., positive LDA weight for PC1 and negative weight for PC2). (j) Reconstructed Ca^{2+} transients for cluster means show that “L” micropatterned cells decayed more slowly than “O” micropatterned cells at 3 μM YODA 1. (k) Illustration of a nonlinear model for Ca^{2+} transients incorporating an exponential rise, a slower exponential decay, and a steady-state constant (left). Representative model fits (median model error) for “O” and “L” micropatterned cells, respectively, at 3 μM YODA 1 (middle). Fitted decay time ($1/\tau_{\text{decay}}$) was significantly longer for “L”-shaped cells at 0.3, 1 and 3 μM YODA 1 (right; Y-axis is truncated to $1/\tau_{\text{decay}} = 1,000$ s) ($n = 4$ biological replicates). Unmarked $P \geq 0.05$, ** $P < 0.01$, *** $P < 0.001$, **** $P < 0.0001$.

6. It will be valuable to plot the frequency of calcium transients with and without yoda 1, rather than the amplitude. The steady state calcium transients are missing in these plots, which is a common phenotype in Gcamp6 based calcium imaging.

As noted above we have added arrows to make clear where we added media (control) or YODA1 in Fig. 2c. These single-celled MCF-7 cells (no cell-cell contacts) do not exhibit spontaneous calcium oscillations, some cells exhibited slow repeated increases in calcium after YODA1 stimulation, but these were very few (see Fig. 2c) and their impact would be reflected in our AUC data and Maximum relative $[Ca^{2+}]_{CYT}$ (0-750 s) and Maximum relative $[Ca^{2+}]_{CYT}$ (750 s-end) analyses (Fig. 2f top and bottom panel).

7. Coming back to the reaction kinetics of yoda1, it will be valuable to image for piezo1 channel using immunofluorescence. I am not convinced that administration of administration of Yoda1 will instantaneously increase the localization and activity of piezo1 channel. Also, doing so may help demonstrate if shape has a role in the localization of piezo1 on the plasma membrane.

We thank the reviewer for this suggestion. The ability of YODA1 to rapidly increase $[Ca^{2+}]_{CYT}$ in MCF-7 cells is shown in PIEZO1 CRISPR experiments (Fig. 4), which are definitive and also our various controls (liquid addition). We agree, that assessing PIEZO1 distribution in L vs. O would be interesting. But at the time when this part of the study was conducted in 2017, there were no satisfactory validated commercially available antibodies for IF or Western Blots. However, the reviewer's comment is insightful, and we have now included this (page 5 lines 223 – 224) as a suggestion for future studies.

“Other potential mechanisms could include ... or spatial distribution of PIEZO1 ...” (page 5 lines 223 – 224).

In conclusion, I think the authors need to revise the manuscript to address the mentioned concerns. This I believe will help strengthen the conclusions drawn by the authors.

We thank the reviewer for their suggestions and the opportunity to improve our manuscript for publication in Communications Biology.

Reviewer #2 (Remarks to the Author):

The manuscript by So et al identifies that changing cell morphology can modify Piezo1 expression and calcium signal that induces features of EPI-MES plasticity. The manuscript is interesting and suitable to the scope of Communications Biology. My main concerns are on the first part of the manuscript where the authors establish the correlation between cell geometry and Piezo1 expression. I also have some concern on the interpretation of the Ca signal and the Yap1 signal.

We thank the reviewer for their positive comments and appreciate the opportunity to clarify our work and improve our manuscript based on their feedback.

1. The main claims are based on two types of cell shapes (O and L) on 1100 μm^2 substrates. It's hard to generalize and claim that cell shape regulates Piezo1 expression, especially that measurements on 700 μm^2 do not seem to follow the same trend as on 1100 μm^2 substrates. More substrate sizes and shapes are needed to support the authors' claim.

We thank the reviewer for their suggestions. We have now included data assessing multiple shapes, which makes it clear why our focus on the elongated triangular morphology with associated changes to the text on page 2 lines 78 – 81, and the addition of Supplementary Fig. 1.

“Using commercially available surface micropatterning (refer to Materials and Methods), we then confined MCF-7 cells to various cellular geometry, and we found that the elongated triangular MCF-7 cells (“L” micropattern; 1,100 μm^2) demonstrated the most pronounced gene expression changes (Supplementary Fig. 1).” (page 2 lines 78 – 81)

Supplementary Fig. 1

919
920
921
922
923
924
925
926

Supplementary Fig. 1: 1,100 μm² “L” micropatterns changes the expression of EPI-MES and Ca²⁺ signalling genes in MCF-7 epithelial like breast cancer cells. Heatmap of expression changes (-ΔΔCt) including EPI-MES markers, Ca²⁺ channels and Ca²⁺ channel regulators and other genes on 1,100 μm² (left) or 700 μm² (right) micropatterned plates. Data represent the average gene expression changes normalized to unpatterned wells (-ΔΔCt) (n = 4 biological replicates, except TRPV1 n = 3 biological replicates). Unmarked P ≥ 0.05, * P < 0.05, ** P < 0.01, *** P < 0.001 (“O” vs “L”).

2. does cell morphology alter local tensions in subcellular regions?

Indeed, Albert et al. have reported that changes in morphologies can alter local tensions in subcellular regions (PMID: 24896113). As the initial experiments (Fig. 1 and Supplementary Fig. 2) were conducted with the primary aim of accessing calcium signaling changes, we did not simultaneously assess local tensions in subcellular regions. However, we have now included in the discussion references to studies showing different subcellular tensions with different micropatterns (page 5, lines 225 – 229).

“As previous studies have found that micropatterning can create localized tension at specific subcellular regions³⁷, future studies could investigate PIEZO1 protein expression and the spatial distribution of PIEZO1 in cells adopting different cellular geometries and their correlation with local tensions at subcellular regions.” (page 5, lines 225 – 229)

3. what are the size of cells when they are not constrained to the patterned substrate? Are the cells stretched or compressed in the authors experiments?

We have now included the size of MCF-7 cells not constrained to the patterned substrate (page 5, lines 215 – 217), which is ~1017 μm² (from n = 448 segmented MCF-7 cells on an unpatterned surface). Hence, the 1,100 μm² pattern provided the most suitable and relevant surface area for our studies.

4. In figure 2, is the difference in Ca influx explained by different Piezo1 expression? If not, what are the molecular reasons that caused the different Ca dynamics? There is also an odd long-term increase of Ca signal on O cells.

We thank the reviewer for their comments. Indeed, we agree that the difference in Ca²⁺ influx could be explained not only by different PIEZO1 expression but also by factors such as the sensitivity of the PIEZO1 channel or other associated calcium signaling pathways. We have now added this to the discussion on page 5 lines 223 – 225, “The augmentation of intracellular Ca²⁺ increases induced by the PIEZO1 activator, YODA 1, in 1,100 μm² “L” cells could be explained by the upregulation of PIEZO1 in these cells. Other potential mechanisms could include the sensitivity, kinetics, or spatial distribution of PIEZO1, or changes in other associated Ca²⁺ signaling pathways.”

We agree that the calcium signals seemed to increase slightly after the initial recovery from the peak in some cells and that this contributes to the sustained increase observed more often in the L shape. We have now expanded and significantly enhanced this section by adding a new panel in Fig. 2f. After YODA1 administration, we followed the calcium transients for individual cells for 30 min (Fig. 2a) and found that most of the calcium responses triggered by YODA1 had recovered to basal levels by 30 min (see Fig. 2c). It is not possible with this throughput and temporal resolution to go beyond 30 min. The longer vs. shorter-term effects are addressed in our use of the Maximum relative [Ca²⁺]_{CYT} (peak is shorter term), area under the curve (AUC) and last data point (which considers long/sustained effects). We have now added a new panel which we believe addresses the reviewer's request to better understand the phenomena by adding Maximum relative [Ca²⁺]_{CYT} (0-750 s) and Maximum relative [Ca²⁺]_{CYT} (750 s-end) (Fig. 2f top and bottom panel).

Fig. 2

848
849
850
851
852
853
854
855
856
857
858
859
860
861
862
863
864
865
866
867
868
869
870
871
872

Fig. 2: Ca^{2+} influx induced by the PIEZO1 activator, YODA 1, in MCF-7 cells is dependent on cellular geometry.

(a) Representative fluorescence images of GCaMP6m-MCF-7 cells on “O” (left) and “L” (right) micropatterns. Scale bar = 50 μm . (b) Line scan (dotted line, scale bar = 20 μm) of YODA 1 (3 μM) induced changes in relative $[\text{Ca}^{2+}]_{\text{CYT}}$ (GCaMP6m fluorescence intensity in arbitrary units shown in pseudocolour; scale bar = 200 s) over time in cells on “O” (top) and “L” micropatterns (bottom). (c) $[\text{Ca}^{2+}]_{\text{CYT}}$ changes in GCaMP6m-MCF-7 cells on “O” and “L” micropatterns with increasing YODA 1 concentrations. $[\text{Ca}^{2+}]_{\text{CYT}}$ changes in ten randomly selected representative cells on “O” (left) and “L” (middle) micropatterns and average of all cells (right). (d) $[\text{Ca}^{2+}]_{\text{CYT}}$ area under the curve (AUC), (e) sustained $[\text{Ca}^{2+}]_{\text{CYT}}$, last data point ($\Delta\text{F}/\text{F}_0$) and (f) maximum relative $[\text{Ca}^{2+}]_{\text{CYT}}$ change ($\Delta\text{F}/\text{F}_0$) for cells on “O” and “L” micropatterns induced by increasing concentrations of YODA 1 (0 – 3 μM). **From 0 – 750 s (top panel) and 750 s – end (bottom panel)**. Data shown represent the mean \pm S.E.M. (g) Top components from PCA captured differences between slow and fast decay. (h) Scatter plots of PC1 and PC2 scores of single cells activated by 3 μM YODA 1. (i) Means of “O” and “L” micropatterned cell clusters were significantly different at 3 μM YODA 1 (left). Linear discriminant analysis (LDA) shows that cluster means were best separated along a direction corresponding to the difference between fast and slow decay (right; i.e., positive LDA weight for PC1 and negative weight for PC2). (j) Reconstructed Ca^{2+} transients for cluster means show that “L” micropatterned cells decayed more slowly than “O” micropatterned cells at 3 μM YODA 1. (k) Illustration of a nonlinear model for Ca^{2+} transients incorporating an exponential rise, a slower exponential decay, and a steady-state constant (left). Representative model fits (median model error) for “O” and “L” micropatterned cells, respectively, at 3 μM YODA 1 (middle). Fitted decay time ($1/\tau_{\text{decay}}$) was significantly longer for “L”-shaped cells at 0.3, 1 and 3 μM YODA 1 (right; Y-axis is truncated to $1/\tau_{\text{decay}} = 1,000$ s) ($n = 4$ biological replicates). Unmarked $P \geq 0.05$, ** $P < 0.01$, *** $P < 0.001$, **** $P < 0.0001$.

5. Is the Piezo1-induced changes in marker expression solely a Ca effect? Can these changes be induced by increasing intracellular Ca²⁺ level while not activating Piezo1?

Calcium signaling indeed affects changes in marker expression (PMID: 23686305, PMID: 33322037), and this is now discussed on page 5 lines 244 – 245 “We and others have shown that Ca²⁺ signaling influences EPI-MES plasticity in breast cancer cells^{29,42”}. In our studies, we show that the changes in marker expression are related to PIEZO1 induced-Ca²⁺ increases as YODA 1-induced activation of PIEZO1 resulted in changes in EMT markers such as SERPINE1. We now provide further evidence for this link via new experiments showing that SERPINE1 levels are very sensitive to calcium and that chelating intracellular Ca²⁺ using BAPTA-AM eliminates SERPINE1 induction by YODA 1 (Supplementary Fig. 4c). This work is also described in page 4 lines 170 – 173 of the new manuscript “SERPINE1 levels were particularly sensitive to intracellular Ca²⁺ levels and its upregulation by YODA 1 was Ca²⁺ signal-dependent as demonstrated by the effects of intracellular Ca²⁺ chelation with BAPTA-AM (Supplementary Fig. 4c)”.

Supplementary Fig. 4

949
950
951
952
953
954
955
956
957
958
959
960

Supplementary Fig. 4: YODA 1-mediated Ca²⁺ influx induces changes in the expression of specific EPI-MES markers.

(a) YODA 1-induced [Ca²⁺]_{CYT} increases are through influx from extracellular Ca²⁺. Media free of extracellular Ca²⁺ (PSS BAPTA) eliminates YODA 1-induced increases in [Ca²⁺]_{CYT} in MDA-MB-468 breast cancer cells. (b) Maximum relative [Ca²⁺]_{CYT} (ΔF/F₀) induced by YODA 1 in media with (PSS Ca²⁺) and without extracellular Ca²⁺ (PSS BAPTA). (c) Assessment of mRNA expression of *CDH1*, *VIM*, *CLDN4* and *SERPINE1* in MDA-MB-468 cells in the presence of YODA 1 with and without BAPTA-AM (6 h). (d) Alterations in mRNA expression of *CDH1*, *VIM*, *CLDN4* and *SERPINE1* in MCF-7 cells induced by YODA 1 (24 h). Where appropriate, data shown represent the mean ± S.E.M. (n = 3 biological replicates). Unmarked P ≥ 0.05, * P < 0.05, ** P < 0.01, *** P < 0.001.

6. In figure 5a, the changes in Yap1 level is very unclear.

Franklin et al. (PMID: 32917893) recently characterized that YAP1 activity does not have a simple linear correlation with its nuclear localization but is dependent on specific nuclear-cytoplasmic exchange. They characterized a localization-reset phenomenon that precedes YAP1 target gene expression. They also found that calcium signaling can trigger the localization-reset of YAP1. In agreement with their findings, we observed the same "initial depletion and subsequent accumulation of nuclear YAP1" (page 4 line 193) in MDA-MB-468 cells after YODA1 (3 μ M) activation. We measured the YAP1 localization-reset at single-cell (Fig. 5b, top and middle panel) and the population level (Fig. 5b bottom panel). We observed the reset at the second time point of YODA1 addition, shown as a dip in nuclear YAP1 in the YODA1 treated cells.

We have now added in the manuscript "Here, we also observed a decrease in nuclear YAP1 upon YODA1 treatment at 15 min and a subsequent "reset" at 1 h time point (Fig. 5a and b)." (page 4 lines 194 – 196)

We have also used dotted lines to indicate the nucleus in Fig. 5a (left panel) to make clearer the nuclear and cytoplasmic regions.

Fig. 5

904 **Fig. 5: YODA 1 induces localization-reset of YAP1 and upregulates YAP1 target gene SERPINE1 in**
 905 **breast cancer cells.**

906 (a) Representative images showing immunofluorescence of YAP1 with dotted lines indicating the nuclei
 907 (left) and nuclei (DAPI, middle) of MDA-MB-468 cells at 0 min (basal), 15 min, 1 h and 3 h following 0 μM
 908 and 3 μM YODA 1 stimulation. Scale bar = 20 μm. Zoomed-in images showing YAP1 localization and the
 909 nucleus of a representative cell (right, scale bar = 10 μm). (b) YODA 1 induced an initial depletion (15
 910 min) and a subsequent accumulation of nuclear YAP1. Nuclear YAP1 changes quantitated as correlation
 911 coefficient, represented as violin plots (top), histograms (middle) and means (bottom). (c) mRNA
 912 expression of *CTGF* and *SERPINE1* in MDA-MB-468 cells after ON-TARGETplus SMARTpool siRNA-
 913 mediated silencing of *YAP1*, *TEAD4* and *WWTR1* (encodes for TAZ). Data shown represent the mean ±
 914 S.E.M. (n = 3 biological replicates). ns P ≥ 0.05, * P < 0.05, ** P < 0.01, *** P < 0.001, **** P < 0.0001.

Reviewer #3 (Remarks to the Author):

In this manuscript, the authors explore the potential role of the mechanosensitive ion channel PIEZO1 in cell shape and plasticity. In addition to investigating the link between Ca²⁺ signalling, cell shape and epithelial to mesenchymal plasticity. Of particular interest is the link between PIEZO1 and SERPINE1 expression and other genes involved in epithelial to mesenchymal plasticity. The role of PIEZO1 in cell morphology is an emerging area and the work presented is novel and of interest to the field. The manuscript overall is very well written and presented.

We thank the reviewer for their positive comments.

Major comments

1. Are protein levels of Piezo1 changed in o versus L shaped cells? please provide western blots of protein expression levels for MCF 7 cells cultured in O or L conformations, if technically feasible.

We thank the reviewer for the comments and for understanding that harvesting proteins from micropatterned cells is technically challenging. Indeed, the yield of protein from each well would be too low, as each well has only ~3000 cells. In addition, at the time when this part of the study was conducted in 2017, there were no appropriate commercially available antibodies for IF or Western Blots. However, the reviewer's comment is definitely valid, and we have now included such studies in our discussion for future work (page 5, lines 227 – 228) "future studies could investigate PIEZO1 protein expression ... in cells adopting different cellular geometries..."

2. Are the channel kinetics affected by changes in cell shape or is the increased calcium influx due to increased protein expression? Please provide evidence using patch clamp.

Patching these cells on these plates at a sufficient number is beyond the scope of this study. However, we now discuss this possible future work in the manuscript (page 5, lines 229 – 231) "Future work could also evaluate the relative contributions of changes in PIEZO1 levels and/or channel kinetics to the shape-dependent increases in calcium influx induced by YODA 1 that have been observed in our studies".

3. Does PIEZO1 knockdown abrogate changes in cell morphology in MCF7 cell cultured in O and L morphologies? The group have used the necessary CRISPR/Cas9 PIEZO1 knockout technology or transient knockout capabilities in this manuscript.

Assessing the ability of PIEZO1 KO to influence the ability of cells to adopt different morphologies (e.g., L vs. O) is not straightforward and was not the aim of this study. It would be hard to determine how a change in % of cells attaching to L patterns might reflect an ability to adopt a shape change vs. a change in attachment efficiency. However, we now cite papers that do report an association with PIEZO1 and cell morphology, as this possible role for PIEZO1 raised by the reviewer is important.

"PIEZO1 is associated with cellular organization and upregulation of PIEZO1 occurs in disorganized malignant T4-2 breast cells compared with non-malignant S1 cells that grow in organized acini³⁸" (page 5 lines 233 – 235)

"Our results show that in addition to being a ... regulator of cellular morphology in CHO- α 4WT cells⁷ and muscle stem cells⁴¹, PIEZO1 can be remodeled as a consequence of changes in cell shape ..." (page 5 lines 240 – 241)

4. Is PIEZO1 sensitivity to mechanical stimulation (as opposed to pharmacological stimulation) affected during changes in cell morphology in the O and L morphologies?

We thank the reviewer for this comment. We, like the reviewer, were also curious if PIEZO1 sensitivity to mechanical stimulation (as opposed to pharmacological stimulation) was affected during changes in cell morphology in the O and L morphologies. However, we did not observe that the L cells were more mechanically sensitive. During compound addition using our high-throughput imaging system, we sometimes noticed a "touch response" in mechanically sensitive cell lines, during which the addition of

liquid (without pharmacological activators) could induce a calcium response (see minor change in Fig. 2c, O chape 0 μM YODA (Media)). However, we did not observe a difference in “touch” sensitivity during the addition of DMSO control in O and L cells and this is reflected in no difference in panels d-f in Fig 2. It would not be possible to use a commercial or custom-built device to dynamically alter cell geometry in a high-throughput calcium imaging device. Such an advance would be a transformative breakthrough in the field and is therefore beyond the scope of this paper.

5. Please confirm changes in PIEZO1 protein expression in MDA-MB-468 cells as well as in knockdowns.

When these experiments were conducted, there were no commercially available, CRISPR-validated PIEZO1 antibodies, and hence we relied on functional calcium influx assays (Fig. 4e). However, as we had retained the samples for this paper, we conducted new experiments with a now commercially available antibody, and we demonstrated PIEZO1 knockdown at the protein level using immunoblotting (Supplementary Fig. 5a). Our genotyping (Methods and Supplementary Table 1) also showed a high knockdown score of ~80 and an indel percentage of ~90%.

We have now added to the manuscript a new figure (Supplementary Fig. 5a), showing the results and relevant information in Materials and Methods “The antibodies used were anti-PIEZO1 antibody (1:1000 in 5% w/v milk PBST, NBP275617, Novus Biologicals)... (Page 9)”

962
963 **Supplementary Fig. 5: The effect of CRISPR-Cas9-mediated *PIEZO1* knockout on basal EPI-MES**
964 **marker expression and YODA 1-mediated protein expression changes.**
965 **(a) Densitometry and representative immunoblot showing knockdown of *PIEZO1* in MDA-MB-468 cells.**
966 **(b) CRISPR-Cas9-mediated *PIEZO1* knockout does not change the basal mRNA expression of *CDH1*,**
967 ***VIM*, *CLDN4* and *SERPINE1* in MDA-MB-468 cells. (c) Densitometry and representative immunoblot**
968 **showing knockdown of *PIEZO1* eliminates the downregulation of the epithelial marker, E-cadherin, in**
969 **MDA-MB-468 by YODA 1 (72 h). (d) Densitometry and representative immunoblot showing knockdown of**
970 ***PIEZO1* changes YODA 1-mediated induction of the mesenchymal marker vimentin in MDA-MB-468 cells**
971 **(24 h). Data shown represent the mean \pm S.E.M. (n = 3 biological replicates). ns P \geq 0.05, * P < 0.05, ****
972 **P < 0.01.**

Minor Comments

6. The work proposes that the increase in PIEZO1 activity is due to increased gene expression over the 16-24 hour culture period. How does this culture time scale compare to physiological relevant changes in cell shape? Please provide more description of the physiological relevance of the O/L pattern culture system.

We thank the reviewer for the comments. During cancer metastasis, cancer cells change their shape as they squeeze through blood vessels. When quantifying in vitro, such as using scratch assay, the migration of cancer cells can occur in hours (PMID: 17406593, PMID: 15576902). A 16–24-hour culture period is sufficient to recapitulate the shape changes that occur during cancer cell migration. Here, we confined breast cancer cells to the non-polarized rounded ("O" micropatterned) pattern, typical of non-migrating cells, and "L", which created a single free-edge organization.

We have now included "Fig. 1b-d shows the comparison between non-polarized rounded ("O" micropatterned) and elongated triangular ("L" micropatterned) geometries (Fig. 1b-d)." (page 2 lines 81 – 82), and "The elongated triangular cellular geometry creates a long single free-edge organization not seen in other micropatterns used in this study." (page 5 lines 217 – 219)

7. Nonomura et al, Friedrich et al and Chuntharpursat-Bon et al, have previously reported changes in VE cadherin/B-actin upon YODA1 treatment in endothelial cells. The significance e-cadherin should be discussed in more detail and how it relates to physiological epithelial remodelling.

We thank the reviewer for referring us to the relevant studies. We have now included this work in our updated discussion (page 5 lines 254 – 259) "One feature of EPI-MES plasticity is the loss or decrease of junctional protein E-cadherin ¹. In MDA-MB-468 breast cancer cells, YODA 1 activation resulted in an increase in vimentin and a decrease in E-cadherin. The association of PIEZO1 and junctional proteins is also seen through the interaction of Piezo1 with PECAM1⁴⁹, and Piezo1-induced remodeling of junctional proteins in endothelial cells and lymphatic endothelial cells ^{50,51}. Future studies could further evaluate the intersection of PIEZO1 and junctional proteins in the context of cancer progression pathways."

8. The interpretation of the results should be in a separate discussion section and should go into more detail for each conclusion.

We thank the reviewer for the comments. We have now included a separate discussion section and have expanded our discussion as suggested (pages 4 – 6).

Reviewer #1 (Remarks to the Author):

I thank the authors for providing explanation/answers to my comments in sufficient detail. I am not convinced by the rebuttal to comment#3 that a cell stretching device is not possible to build or buy off the shelf. But the authors have astutely removed the word 'stretch' from the text to present the data in a different context. However, given the scope of the paper and the journal, I will concede that the new experiments and modified figures and associated texts have improved the manuscript significantly. Therefore, I recommend this manuscript for publication.

We thank the reviewer for their positive comments.

Reviewer #3 (Remarks to the Author):

The authors have addressed all concerns. Where technically possible they have included new data and otherwise provided further detail in the text. They have expanded the experimental detail and interpretation of their data. I would like to thank the authors for an elegant manuscript and their effort in addressing the extensive review comments.

We thank the reviewer for their positive comments.